# SUBWORDS AS SKILLS: TOKENIZATION FOR SPARSE-REWARD REINFORCEMENT LEARNING

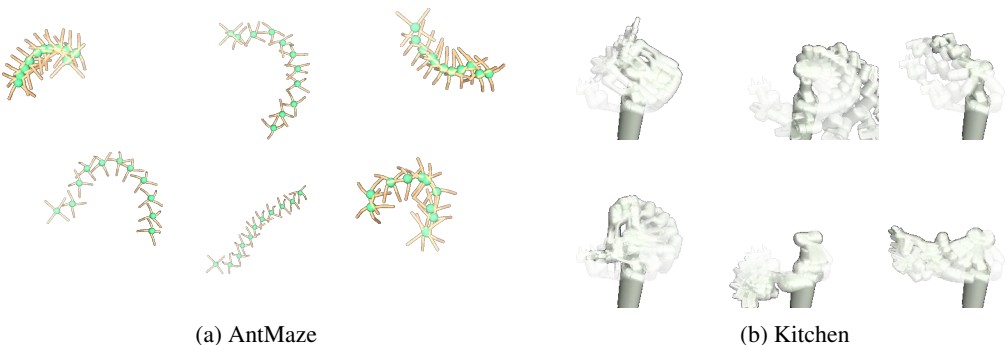

(a) AntMaze           (b) Kitchen

Figure 1: A sample of some "skills" that our method identifies for the (a) AntMaze and (b) Kitchen environments, where the transparency is higher (color is paler) for poses earlier in the trajectory. For more discussion see Appendix B.

## ABSTRACT

Exploration in sparse-reward reinforcement learning (RL) is difficult due to the need for long, coordinated sequences of actions in order to achieve any reward. Moreover, in continuous action spaces there are an infinite number of possible actions, which only increases the difficulty of exploration. One class of methods designed to address these issues forms temporally extended actions, often called skills, from interaction data collected in the same domain, and optimizes a policy on top of this new action space. Such methods require a lengthy pretraining phase in order to form the skills before reinforcement learning can begin. Given prior evidence that the full range of the continuous action space is not required in such tasks, we propose a novel approach to skill-generation with two components. First we discretize the action space through clustering, and second we leverage a tokenization technique borrowed from natural language processing to generate temporally extended actions. Using this as an action-space for RL outperforms comparable skill-based approaches in several challenging sparse-reward domains, and requires orders-of-magnitude less computation.

## 1 INTRODUCTION

Reinforcement learning (RL), the learning paradigm that allows an agent to interact with an environment and collect its own data, is a promising approach to learning in many domains where high-quality data collection is too financially expensive or otherwise intractable. Though it began with dynamic programming in tabular settings, the recent use of neural networks as function approximators has led to great success on many challenging learning tasks (Mnih et al., 2013; Silver et al., 2017; Gu et al., 2017). These successful tasks tend to have some particular properties. In some cases, it is simple to define a reward function that yields reward at every step of interaction (the "dense" reward setting), like directional velocity of a robot learning to walk (Haarnoja et al., 2018a). In other cases, the environment dynamics are known, as in the case of Chess or Go (Silver et al., 2017). However, for many natural tasks like teaching a robot to make an omelet, it is much more straightforward to tell when the task is completed without knowing how to automatically supervise each individual step,

how to model the environment dynamics. Learning in these "sparse" reward settings, where reward is only obtained extremely infrequently (e.g., at the end of successful episodes) is notoriously difficult. In order for a learning agent to improve its policy, the agent needs to explore its environment for long periods of time, often in a coordinated fashion, until it finds any reward.

One class of solutions to this problem involves including additional task-agnostic dense rewards as bonuses that encourage agents to explore the state space (Pathak et al., 2017; Burda et al., 2018b). Another class of solutions to the exploration issue is to jumpstart the function approximator to be used in reinforcement learning by training it on some pretext task (Yarats et al., 2021; Liu and Abbeel, 2021), which works well when the training and downstream domains are well aligned. A third class of methods aims to create temporally extended actions, or "skills", from interactions or data. A particular subclass of methods learns skills that are conditioned on the observations (Singh et al., 2020; Pertsch et al., 2021; Ajay et al., 2020; Sharma et al., 2019; Eysenbach et al., 2018; Park et al., 2022; 2023), which means that the deployment scenario needs to match the data. Others relax this assumption (Lynch et al., 2020; Pertsch et al., 2021; Bagatella et al., 2022) so that such skills can easily be transferred to some new domain as long as the action space remains the same. This has the potential to speed up exploration in new tasks for which it is not easy to collect data a priori (i.e., few-shot), which can lead to faster task adaptation. However, these recent efforts in skill learning all require lengthy pretraining phases due to their reliance on neural networks in order to learn the skills. Inspired by the recent cross-pollination of natural language processing (NLP) techniques in offline RL (Chen et al., 2021; Janner et al., 2021; Shafiullah et al., 2022), we take a different approach.

Like the long-range coordination required for exploration in sparse-reward RL, language models must model long range dependencies between discrete tokens. Character inputs leads to extremely long sequences, and requires language models to both spell correctly and model inter-word relations. On the other hand, word-level input results in the model poorly capturing certain rare and unseen words. The solution is to create "subword" tokens somewhere in between individual characters and words that can express any text (Gage, 1994; Sennrich et al., 2015; Provilkov et al., 2020; Kudo, 2018; Schuster and Nakajima, 2012; He et al., 2020).

In the spirit of this development in language modeling, we propose a tokenization method for learning skills. Following prior work (Dadashi et al., 2022; Shafiullah et al., 2022), we discretize the action space and use a modified byte-pair encoding (BPE) scheme (Gage, 1994; Sennrich et al., 2015) to obtain temporally extended actions. Then, we use this as the action-space for RL. As we demonstrate, such a method benefits from extremely fast skill-generation (minutes v.s. hours for neural network-based methods), significantly faster rollouts and training due to open-loop subword execution that does not require an additional neural network, interpretability of a finite set of skills, and strong results in several sparse-reward domains.

## 2 RELATED WORK

**Exploration in RL:** Exploration is a fundamental problem in RL, particularly when reward is sparse. A common approach to encouraging exploratory behavior is to augment the (sparse) environment reward with a dense bonus term that biases toward exploration. This includes the use of state visitation counts (Poupart et al., 2006; Lopes et al., 2012; Bellemare et al., 2016) and state entropy objectives (Mohamed and Jimenez Rezende, 2015; Hazan et al., 2019; Lee et al., 2019; Pitis et al., 2020; Liu and Abbeel, 2021; Yarats et al., 2021) that incentivize the agent to reach "novel" states. Related, "curiosity"-based exploration bonuses encourage the agent to take actions in states where the effect is difficult to predict using a learned forward (Schmidhuber, 1991; Chentanez et al., 2004; Stadie et al., 2015; Pathak et al., 2017; Achiam and Sastry, 2017; Burda et al., 2018a) or inverse (Haber et al., 2018) dynamics model. Burda et al. (2018b) propose a random network distillation exploration bonus based upon the error in observation features predicted by a randomly initialized neural network.

**Temporally Extended Actions and Hierarchical RL:** Another long line of work explores temporally extended actions due to the potential for such abstractions to improve learning efficiency. These advantages are particularly pronounced for difficult learning problems including sparse reward tasks, which is the focus of our work. In particular, action abstractions enable more effective exploration (Nachum et al., 2018) and simplify the credit assignment problem. Hierarchical reinforcement learning (HRL) (Dayan and Hinton, 1992; Kaelbling, 1993; Sutton, 1995; Boutilier et al., 1997; Parr and Russell, 1997; Parr, 1998; Sutton et al., 1999; Dietterich, 2000; Barto and Mahadevan, 2003;

Kulkarni et al., 2016; Bacon et al., 2017; Vezhnevets et al., 2017) considers the problem of learning policies with successively higher levels of abstraction (typically two), whereby the lowest level considers actions directly applied in the environment while the higher levels reason over temporally extended transitions. A classic example of action abstractions is the options framework (Sutton et al., 1999) , which provides a standardization of HRL in which an option is a terminating sub-policy that maps states (or observations) to low-level actions. Options are often either prescribed as predefined low-level controllers or learned via subgoals or explicit intermediate rewards (Dayan and Hinton, 1992; Dietterich, 2000; Sutton et al., 1999). Some simple instantiations of options include repeated actions (Sharma et al., 2017) and self-avoiding random walks (Amin et al., 2020). Konidaris and Barto (2009) learn a two-level hierarchy by incrementally chaining options ("skills") backwards from the goal state to the start state. Nachum et al. (2018) propose a hierarchical learning algorithm (HIRO) that learns in an off-policy fashion and, in turn, is more sample-efficient than typical HRL algorithms, which learn on-policy. Achieving these sample efficiency gains requires addressing the instability typical of off-policy learning, which is complicated by the non-stationarity that comes with jointly learning low- and high-level policies. Levy et al. (2017) use different forms of hindsight (Andrychow-icz et al., 2017) to address similar instability issues that arise when learning policies at multiple levels in parallel.

**Skill Learning from Demonstrations:** In addition to the methods mentioned above in the context of HRL, there is an existing body of work that seeks to discover extended actions prior to their use in online RL, often called "skills". Many methods have been developed for skill discovery from interaction (Daniel et al., 2012; Gregor et al., 2016; Eysenbach et al., 2018; Warde-Farley et al., 2018; Park et al., 2022; 2023). Most related to our setting is a line of work that explores extended action discovery from demonstration data (Lynch et al., 2020; Ajay et al., 2020; Singh et al., 2020; Pertsch et al., 2021; Bagatella et al., 2022). As an example, Lynch et al. (2020) learn a VAE on chunks of action sequences in order to generate a temporally extended action by sampling a single vector. Ajay et al. (2020) follow a similar approach, but use flow models on top of entire trajectories, and only rollout a partial trajectory at inference time. Some of these methods (Ajay et al., 2020; Singh et al., 2020; Pertsch et al., 2021) condition on the observations when learning skills, which leads to more efficient exploration, but such conditioning means that any skill that is learned will need to be deployed in the same environment as the one in which the data was collected, resulting in poor domain transfer performance (Bagatella et al., 2022). Others (Lynch et al., 2020; Bagatella et al., 2022) simply condition on actions, which means that the skills can be reused in any domain that shares the same action space. In an effort to learn more generalizable skills, we follow this latter example. There is also a related prior work that applies grammar-learning to online RL (Lange and Faisal, 2019), but such a method learns an ever-growing number of longer actions, which poses significant issues in the sparse-reward setting, as we discuss later.

## 3 METHOD

Similar to prior work (Lynch et al., 2020; Ajay et al., 2020; Singh et al., 2020; Pertsch et al., 2021; Bagatella et al., 2022), we extract skills from demonstration data, more formally a dataset of $N$ trajectories with lengths $\{n_i\}_{i \in N}$ that involve the same action space as our downstream task

$$\mathcal{D} = \left\{ (o_{ij}, a_{ij})_i | i \in \mathbb{N} \cap [0, N), \ j \in \mathbb{N} \cap [0, n_i), \ o_{ij} \in \mathbb{R}^{d_{\text{obs}}}, \ a_{ij} \in \mathbb{R}^{d_{\text{act}}} \right\},$$

where $a_{ij}$ and $o_{ij}$ denote actions and observations, respectively. After extracting skills from this dataset, we use these skills as a new action space for reinforcement learning on some downstream task. Crucially, our skills are unconditional so do not have any information as to when they should be used in the downstream task. In following sections we detail our exact method.

### 3.1 BYTE-PAIR ENCODING

Byte-pair encoding (BPE) was first proposed as a simple method to compress files (Gage, 1994), but it has recently been used to construct vocabularies for NLP tasks in between the resolution of characters and whole-words (Sennrich et al., 2015). With character vocabularies, the vocabulary is small, but the sequence lengths are large. Such long sequences are extremely burdensome to process, especially for the current generation of Transformers. In addition, making predictions at the character level imposes a more difficult task on the language model: it needs to spell everything correctly,

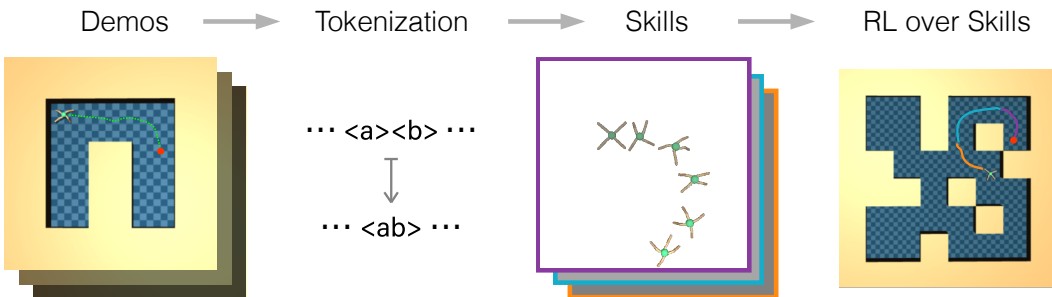

Figure 2: Abstract representation of our method. Given demonstrations in the same action space as our downstream task, we discretize the actions and apply tokenization techniques to recover "subwords" that form a vocabulary of skills. We then train a policy on top of these skills for some new task. We only require a common action space between demonstrations and downstream task.

or make a long-coordinated set of predictions, not unlike the requirement on action sequences for sparse-reward exploration. Whole-word vocabularies shorten the sequence lengths and make the prediction task easier, but if a word is rare or, even worse, unseen in the training data, the outputs of the language model may not be correct in many cases. Subword vocabularies have emerged as a sweet-spot between these two extremes and are widely used in language models (Schuster and Nakajima, 2012; Sennrich et al., 2015; Kudo, 2018; Provilkov et al., 2020; He et al., 2020).

Given a long sequence of tokens and an initial fixed vocabulary, BPE consists of two core operations: (i) compute the most frequent pair of neighboring tokens and add it to the vocabulary, and (ii) merge all instances of the pair in the sequence. These two steps of adding tokens and making merges alternate until a fixed maximum vocabulary size is reached.

### 3.2    DISCRETIZING THE ACTION SPACE

In order to run BPE, it is necessary to have an initial vocabulary $\mathcal{V}$ as well as a string of discrete tokens. In a continuous action space, one simple way to form tokens is through clustering. Prior work has leveraged these ideas in similar contexts (Janner et al., 2021; Shafiullah et al., 2022; Jiang et al., 2022) and we follow suit. For simplicity, we perform $k$-means clustering with the Euclidean metric on the actions of demonstrations in $\mathcal{D}$ to form a vocabulary of $k$ discrete tokens $\mathcal{V} = \{v_0, \ldots, v_k\}$. Our default choice for $k$ will be two times the number of degrees-of-freedom (DoF) of the original action space, or $2 \cdot d_{\text{act}}$. We will further study this choice in Appendix A.1. Such a clustering is the same as the action space of Shafiullah et al. (2022) without the residual correction.

### 3.3    SCORING MERGES

In NLP, we often have access to a large amount of text data from (mostly) correct human authors. However, for robotics applications we may not have the same quantity of near-optimal (or even suboptimal) demonstrations. As a result, it may be undesirable to merge tokens based on frequency alone. Thus, in addition to merging based on frequency, we implement a variant of our method that merges based on a proxy for the distance traveled in the observation space in order to encourage the creation of skills that explore diversely in state space and thus are efficient for tasks. We take inspiration from LSD (Park et al., 2022) and CSD (Park et al., 2023) for this choice. At the high sampling rate of continuous control observations, the observation space should be locally Euclidean, so such a measure makes sense as long as the length of skills is short enough. We label the two variants of our method as **SaS-freq** and **SaS-dist** respectively (SaS for Subwords as Skills).

More formally, suppose that two neighboring subwords $w_1$ and $w_2$ correspond to the trajectories $\tau_1 = \{(o_1, a_1), \ldots, (o_n, a_n)\}$ and $\tau_2 = \{(o_{n+1}, a_{n+1}), \ldots, (o_m, a_m)\}$. For an instance of the subword $w = \text{concat}(w_1, w_2)$ consisting of the entire trajectory $\tau = \text{concat}(\tau_1, \tau_2)$, we associate the vector $q_\tau = \frac{1}{m} \sum_{i=1}^{m} (o_i - o_1)$. This vector is analogous to the average "heading" of the subword, which ignores possible high-frequency, periodic motion like legs moving up and down. In order to obtain a vector that summarizes $w$, we compute the mean of such instances $q_w = \mathbb{E}_{(\tau_1, \tau_2) \in \mathcal{D}}[q_\tau]$, which takes into account possible observation noise at different instances.

---

**Algorithm 1** Subword merging and pruning

---

1: Given dataset $\mathcal{D} = \{(o_{ij}, a_{ij})_i | i \in \mathbb{N} \cap [0, N), \ j \in \mathbb{N} \cap [0, n_i), \ o_{ij} \in \mathbb{R}^{d_{\text{obs}}}, \ a_{ij} \in \mathbb{R}^{d_{\text{act}}}\}$
2: Given $k, N_{\max}, N_{\min}, \epsilon \ll 1$
3: Run $k$-means on actions with $k$ clusters to get tokens $\mathcal{V} = \{v_i\}_{i=1}^{k}$
4: Tokenize $\mathcal{D}$ according to $\mathcal{V}$
5: Initialize $\mathcal{W} = \{w_i\}_{i=1}^{k} \leftarrow \mathcal{V}, \mathcal{Q} \leftarrow \varnothing, \bar{q} = 0, \Sigma_q = I$
6:                                                                        // Merge vocabulary
7: **while** $|\mathcal{W}| < N_{\max}$ **do**
8:    $\mathcal{W}' \leftarrow \{\text{All possible merges } w = \text{concat}(w_1, w_2) \text{ in } \mathcal{D} \mid w_1, w_2 \in \mathcal{W}\}$    // Get candidates
9:    **for** $w' \in \mathcal{W}'$ **do**
10:       Compute $q_{w'} = \mathbb{E}_{\text{instances of } w' \text{ in } \mathcal{D}} \left[ \frac{1}{L} \sum_{r=1}^{L \ = \ \text{length of } w'} o_{ir} - o_{i1} \right]$    // Compute vectors
11:    **end for**
12:    $w' = \arg\max_{w' \in \mathcal{W}'} (q_{w'} - \bar{q})^{\top} \Sigma_q^{-1} (q_{w'} - \bar{q})$    // Find best possible merge
13:    $\mathcal{W} \leftarrow \mathcal{W} \cup \{w'\}, \mathcal{Q} \leftarrow \mathcal{Q} \cup \{q_{w'}\}$    // Add merge to vocabulary
14:    $\bar{q} \leftarrow \mathbb{E}_{q \in \mathcal{Q}}[q], \Sigma_q \leftarrow \text{Cov}_{q \in \mathcal{Q}}(q) + \epsilon I$    // Update vocabulary mean and covariance
15:    Retokenize $\mathcal{D}$ according to $\mathcal{W}$
16: **end while**
17:                                                                        // Prune vocabulary
18: **while** $|\mathcal{W}| > N_{\min}$ **do**
19:    $w' = \arg\min_{w' \in \mathcal{W}} (q_{w'} - \bar{q})^{\top} \Sigma_q^{-1} (q_{w'} - \bar{q})$    // Find most redundant subword
20:    $\mathcal{W} \leftarrow \mathcal{W} \setminus \{w'\}, \mathcal{Q} \leftarrow \mathcal{Q} \setminus \{q_{w'}\}$    // Remove worst
21:    $\bar{q} \leftarrow \mathbb{E}_{q \in \mathcal{Q}}[q], \Sigma_q \leftarrow \text{Cov}_{q \in \mathcal{Q}}(q) + \epsilon I$    // Update vocabulary mean and covariance
22: **end while**
23:
24: **return** $\mathcal{W}$

---

Given an existing vocabulary of subwords $\mathcal{W} = \{w_0, \dots, w_{n-1}\}$ and their corresponding vectors $\mathcal{Q} = \{q_0, \dots, q_{n-1}\}$, we can compute the mean $\bar{q} = \mathbb{E}_{q \in \mathcal{Q}}[q]$ and covariance matrix $\Sigma_q = \text{Cov}_{q \in \mathcal{Q}}(q) + \epsilon I$ for some small $\epsilon$. Now, we associate a score to each possible new subword according to the Mahalanobis distance between the candidate subword and the set of existing subwords: $d_w = (q_w - \bar{q})^{\top} \Sigma_q^{-1} (q_w - \bar{q})$. We add the subword with maximum distance $d_w$ to our vocabulary. We update $\Sigma_q$ and $\bar{q}$ at every iteration. This results in a growing vocabulary of subwords that not only achieve high distance in observation space but are diverse. Such a scoring function also accounts for the fact that different parts of the observation space may have different natural scales. We merge up to a maximum vocabulary size $|\mathcal{W}| = N_{\max}$. The choice of $N_{\max}$ is further studied in Appendix A.2.

### 3.4 PRUNING THE SUBWORDS

If we stopped after merging to a maximum size, the final vocabulary would contain the intermediate subwords that make up the longest units. In the context of NLP, this redundancy may not be particularly detrimental. In reinforcement learning, however, redundancy in the action space of a new policy will result in similar actions competing for probability mass, making exploration and optimization more difficult. Thus we propose pruning the vocabulary.

For frequency-based merging, we start with the longest subword, and remove subwords that are strictly contained in it, then move to the next longest and repeat the process. We do this until we reach the desired vocabulary size $N_{\min}$.

For distance-based merging, we prune the set of subwords using the same metric as was used to merge. In particular, we find $w' = \arg\min_w d_w$, update $\mathcal{W} \leftarrow \mathcal{W} \setminus \{w'\}$, and recompute $\Sigma_q$ and $\bar{q}$. We continue pruning in this fashion until reaching a minimum vocabulary size $|\mathcal{W}| = N_{\min}$. Finally, $\mathcal{W}$ becomes the action space for a new policy. Algorithm 1 provides the pseudocode for the distance-based method, and Figure 2 provides a graphical representation. We ablate the choice of $N_{\min}$ in Appendix A.3.

Implicit in our method is an assumption that portions of the demonstrations can be recomposed to solve a new task, i.e., that there exists a policy that solves the new task with this new action space.

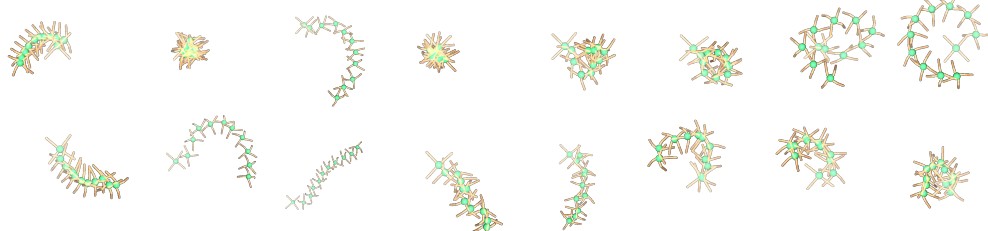

Figure 3: All skills generated for `antmaze-medium-diverse` where the transparency is higher for poses earlier in the trajectory. See Appendix B for more details.

One can imagine a counter-example where the subwords we obtain lack some critical action sequence without which the task cannot be solved. Still, we will show that this is a reasonable assumption for several sparse-reward tasks.

## 4 EXPERIMENTS

In the following sections, we explore the empirical performance of our proposed method: first extracting skills from data, then using those skills as an action space for learning a new policy through sparse-reward RL. We see that there are significant speed and performance benefits, with strong exploration behavior. We also discuss benefits and drawbacks of our unconditional skills when compared to conditional skills like those of SPiRL (Pertsch et al., 2021).

### 4.1 REINFORCEMENT LEARNING WITH UNCONDITIONAL SKILLS

Table 1: Main comparison (unnormalized scores). SSP corresponds to results from official code of Pertsch et al. (2021). We report numbers at the end of training for consistency. SFP takes so long it is unmanageable on many domains. AntMaze is scored 0–1, Kitchen is scored 0–4 in increments of 1, CoinRun is scored 0–100 in increments of 10. *CoinRun is a discrete-action domain, so instead of SAC only SAC-discrete can be used. SSP results exist for Kitchen, ($0.8_{\pm0.2}$ (Pertsch et al., 2021, Figure 4)), but we are unable to reproduce this number using official code.

| Task | SAC | SAC-discrete | SSP | SFP | SaS-freq | SaS-dist |
|---|---|---|---|---|---|---|
| `antmaze-umaze-diverse` | 0.0 | 0.0 | 0.0 | — | 0.0 | **0.76**$_{\pm\textbf{0.43}}$ |
| `antmaze-medium-diverse` | 0.0 | 0.0 | 0.0 | 0.0 | 0.0 | **0.40**$_{\pm\textbf{0.55}}$ |
| `antmaze-large-diverse` | 0.0 | 0.0 | 0.0 | 0.0 | 0.0 | **0.34**$_{\pm\textbf{0.46}}$ |
| `kitchen-mixed` | 0.0 | 0.0 | 0.0* | $0.12_{\pm0.07}$ | $0.16_{\pm0.17}$ | **0.72**$_{\pm\textbf{0.40}}$ |
| `CoinRun` | —* | 0.0 | **5.3**$_{\pm\textbf{3.4}}$ | 0.0 | $4.90_{\pm9.10}$ | $2.9_{\pm2.9}$ |

**Tasks:** We consider AntMaze and Kitchen from D4RL (Fu et al., 2020), two challenging sparse-reward state-based tasks/datasets. AntMaze is a maze navigation task with a quadrupedal robot where the reward is 0 except for at the goal, and Kitchen is a manipulation task in a kitchen setting where reward is 0 except for on successful completion of a subtask. Demonstrations in AntMaze consist of random start and end states in the same maze collected by a suboptimal scripted policy, while demonstrations in Kitchen consist of different sequences of subtasks than the eventual goal in the same kitchen collected by humans in VR. We also consider CoinRun (Cobbe et al., 2019), a discrete-action platforming game. Unlike AntMaze and Kitchen, CoinRun is a visual domain and the demonstrations are collected by humans in distinct levels than the final task. All of these domains require many coordinated actions in sequence to achieve any reward, with horizons between 280 and 1000 steps. See Appendix E for more information on the data.

**Baselines:** We consider SAC (Haarnoja et al., 2018b); SAC-discrete (Christodoulou, 2019) on top of our discretized $k$-means actions; Skill-Space Policy (SSP), a VAE trained on sequences of 10 actions at a time (Pertsch et al., 2021); and State-Free Priors (SFP) (Bagatella et al., 2022) a sequence model of actions that is used to inform action-selection during SAC inference, which takes the last action

as context. For SAC, SAC-discrete, SSP, and SFP, we implement or run the official code with the default hyperparameters listed in the respective papers. Complete results are available in Table 1. All numbers are taken from the end of training. We report mean and standard deviation across five seeds. As defaults we use $k = 2 \cdot d_{act}$ and $N_{min} = 16$. We pick $N_{max}$ per-domain such that skill lengths are comparable with SSP's length 10. For more experimental details see Appendix E. Including our method, all skills are not conditioned on observations.

We see in Table 1, that even in these challenging sparse-reward tasks, our method is the only one that is able to achieve nonzero reward across all tasks. All settings with zero reward fail to achieve any reward during training. The large standard deviations are due to the fact that some seeds fail to achieve any reward. Figure 3 visualizes 200-step rollouts of all of the discovered subwords for

Table 2: Per-domain subword lengths. Numbers are intended to match the length-10 skills of SSP, but it is difficult to precisely control length due to the merging and pruning process.

| Task | Subword length |
|---|---|
| antmaze-umaze-diverse | $11.3_{\pm 5.6}$ |
| antmaze-medium-diverse | $8.5_{\pm 5.0}$ |
| antmaze-large-diverse | $12.5_{\pm 5.3}$ |
| kitchen-mixed | $9.2_{\pm 4.5}$ |
| CoinRun | $9.1_{\pm 5.6}$ |

`antmaze-medium-diverse`. We provide mean and standard deviations for subword lengths in extracted vocabularies in Table 2. Failures of frequency-based merging in AntMazes are directly attributable to the discovery of long, constant sequences of actions, likely due to suboptimal demonstration trajectories that often jitter in place.

Due to the simplicity of our method, it also enjoys significant acceleration compared to the baselines. In Table 3, we measure the wall-clock time required to generate skills, as well as inference for a single rollout. We see that our method achieves extremely significant speedups compared to prior work, achieving both faster and more efficient learning, as well as faster inference during execution. Our skill discovery is fast as we simply need to run $k$-means and tokenization. SSP and SFP require training larger generative models. In

Table 3: Timing on `antmaze-medium-diverse` in seconds. Methods measured on the same Nvidia RTX 3090 GPU with 8 Intel Core i7-9700 3 GHz CPUs @ 3.00 GHz. SSP takes around 36 hours for skill generation and SFP takes around 2 hours.

| Method | Skill Generation | Online Rollout |
|---|---|---|
| SSP | $130000_{\pm 1800}$ | $0.9_{\pm 0.05}$ |
| SFP | $8000_{\pm 500}$ | $4.1_{\pm 0.1}$ |
| SaS-dist | $\mathbf{210}_{\pm 10}$ | $\mathbf{0.007}_{\pm 0.0006}$ |

the case of rollouts our method predicts an entire sequence of actions using a simple policy every 10 steps or so, while SSP and SFP require much larger models in order to predict the latent variable, and then generate the next action from that latent. The speedup of our method also translates to faster RL (around 10 hours for our method vs. 12 hours for SSP and 1 week for SFP), which leads to faster iteration.

## 4.2 EXPLORATION BEHAVIOR ON ANTMAZE MEDIUM

The stringent evaluation procedure for sparse-reward RL equally penalizes poor learning and exploration. In order to shed light on the many zeros in Table 1, we examine the exploration behavior on AntMaze Medium. We choose this domain because it is particularly straightforward to interpret what good and bad exploration looks like: coverage of the maze. In Figure 4 and Figure 5 we plot state visitation for the first 1 million of 10 million steps of RL. We show the approximate start position in grey in the bottom left and the approximate goal location in green in the top right. Higher color intensity (saturation) corresponds to a higher probability of that state. Color is scaled nonlinearly according to a power law between 0 and 1 for illustration purposes. Thin white areas between the density and the walls can be attributed to the fact that we plot the center body position, and the legs have a nontrivial size limiting the proximity to the wall.

In Figure 4, we show the exploration behavior across methods, averaged over 5 seeds. We see that the 0 values for the final reward in Table 1 for SAC, SSP and SFP are likely due not to poor optimization, but rather poor exploration early in training, unlike our method. One reason for this could be due to the fact that our subwords are a discrete set, so policy exploration does not include small differences in a continuous space. In addition, SAC has fundamental issues in sparse-reward environments as the

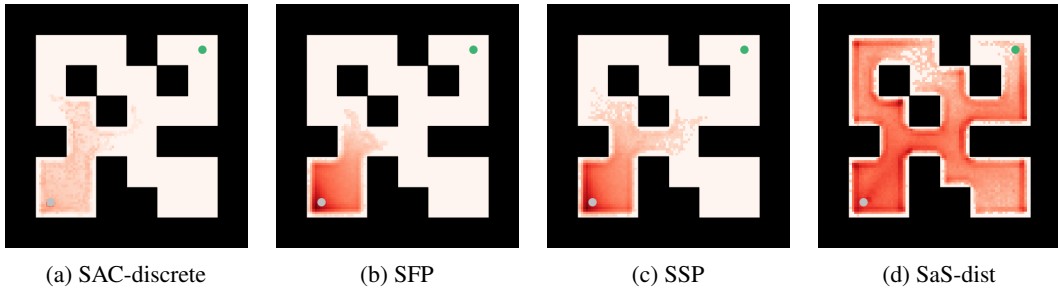

|   (a) SAC-discrete   |   (b) SFP   |   (c) SSP   |   (d) SaS-dist   |

Figure 4: A visualization of state visitation for RL on `antmaze-medium-diverse` in the first 1 million timesteps for (a) SAC-discrete, (b) SFP, (c) SSP, and (d) our method. The grey circle in the bottom-left denotes the start position, while the green circle in the top-right indicates the goal. Notice that our method explores the maze much more extensively. SAC's visitation is tightly concentrated on the start state, which is why there is so little red in (a).

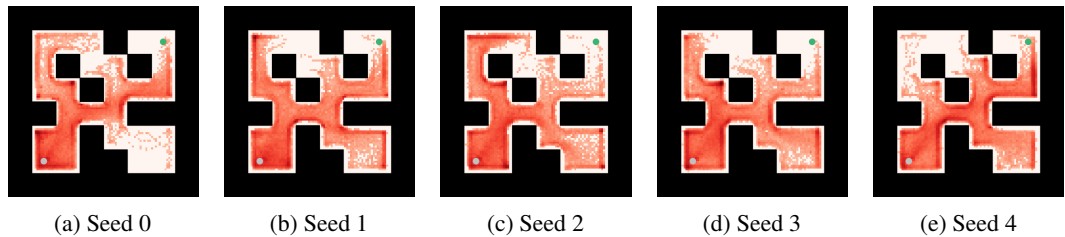

|   (a) Seed 0   |   (b) Seed 1   |   (c) Seed 2   |   (d) Seed 3   |   (e) Seed 4   |

Figure 5: State visitation achieved with our method for each of the 5 individual seeds. Notice the diversity of exploration behavior. This is true even for seeds 0, 2 and 3 that, as reflected in the standard deviations in Table 1, eventually finish with a final reward of 0.

signal to the Q function is driven entirely by the entropy bonus, which will lead to uniform weighting on every action and as a result Brownian motion in the action space. Such behavior is likely why the default setting for SAC (Haarnoja et al., 2018b) aggressively drives the policy to determinism, but in the sparse reward setting this also results in a uniform policy. Without long sequences of coordinated actions such exploration is insufficient.

In Figure 5, we show the individual seed visitation of our method in the first 1 million steps. This is to demonstrate that, even though individual seeds may have some bias, they all are able to explore much more widely than the collective exploration of baseline methods. Indeed, this suggests that the large standard deviations of our method are a result of an optimization failure, as suggested by Zhou et al. (2022), and not poor exploration due to bad skill-encoding.

### 4.3 COMPARISON TO OBSERVATION-CONDITIONED SKILLS

Our method for extracting skills is an unconditional, open-loop method with the idea in mind that the skills should generalize. Still, this comes with the drawback that a policy will have to learn the right context to deploy skills from scratch. Alternatively, observation-conditioned skills bias policy exploration to match that of the demonstrations. This allows for more stable exploration, but worse generalization (Bagatella et al., 2022).

**Baselines:** Here we compare to observation-conditioned extension of SSP, SPiRL and SPiRL-cl (the closed-loop version) (Pertsch et al., 2021; 2022) which bias a policy toward skills used in the exact context of demonstrations in the dataset. We also include OPAL (Ajay et al., 2020), a similar method to SPiRL, a flow model for entire trajectories of actions that is conditioned on observations. We take numbers from the paper as OPAL is closed-source.

In Table 4, we see that SPiRL and SPiRL-cl show very strong performance on Kitchen, where the overlap between the dataset and the downstream task is exact, but SPiRL fails on AntMaze-large, while SPiRL-cl fails on CoinRun, likely due to differences between the dataset for CoinRun (easy levels) and the downstream task (hard levels). In addition we notice that BPE with simple frequency

Table 4: Comparison to methods with observation-conditioned skills. In general we see conditioning helps when the data closely overlaps with the downstream task (Kitchen), but not in CoinRun where such an overlap cannot be assumed. With AntMaze the results are mixed likely due to the suboptimal quality of the demonstrations. We highlight that, even without conditioning, our method is competitive in AntMaze-large and comparable to SPiRL in AntMaze-medium. OPAL is a closed-source method similar to SPiRL, and results are from Ajay et al. (2020).

| Task | SPiRL | SPiRL-cl | OPAL | SaS-freq | SaS-dist |
|---|---|---|---|---|---|
| `antmaze-medium-diverse` | $0.40_{\pm 0.49}$ | $1.00_{\pm 0.00}$ | $0.82_{\pm 0.04}$ | $0.0$ | $0.40_{\pm 0.55}$ |
| `antmaze-large-diverse` | $0.0$ | $0.20_{\pm 0.40}$ | $0.0$ | $0.0$ | $0.34_{\pm 0.46}$ |
| `kitchen-mixed` | $1.87_{\pm 0.16}$ | $3.00_{\pm 0.00}$ | — | $0.16_{\pm 0.17}$ | $0.72_{\pm 0.40}$ |
| CoinRun | $5.32_{\pm 5.41}$ | $0.0$ | — | $4.90_{\pm 9.10}$ | $2.90_{\pm 2.90}$ |

merging (SaS-freq) is poor in AntMaze as discussed previously but comparable in CoinRun. Note that we are able to replicate results for SPiRL-cl (2–3 in the original paper (Pertsch et al., 2022)), but for SPiRL our result is significantly worse (2–3 in the original paper (Pertsch et al., 2021)). It is unclear from where this discrepancy stems, but we use the official code, for which Kitchen is already implemented.

In addition, we examine generalization behavior across observation-conditioned methods. Table 5 highlights the drawback that conditioning has in generalization. In particular the strongest advantage for conditional skills is in a setting where the data closely matches the final task, but it may be detrimental when we do not have access to sufficiently general demonstrations, like the $\sim$10,000 trajectories in randomized environments that SPiRL uses for visual PointMaze (Pertsch et al., 2021).

Table 5: Results on transferring skills extracted from `antmaze-medium-diverse` to downstream RL on `antmaze-umaze-diverse`. We see that methods with conditioning (SPiRL and SPiRL-cl) underperform our simple unconditional method. Similar conclusions were drawn by the authors of SFP (Bagatella et al., 2022, Figures 7, 16), where stronger conditioning fails to generalize.

| Task | SSP | SPiRL | SPiRL-cl | SaS-dist |
|---|---|---|---|---|
| `antmaze-medium-diverse` $\rightarrow$ `antmaze-umaze-diverse` | $0.0$ | $0.60_{\pm 0.49}$ | $0.20_{\pm 0.40}$ | $\mathbf{0.97_{\pm 0.12}}$ |

## 5 CONCLUSION

**Limitations:** As proposed, there are a few key limitations to our method. Discretization removes resolution from the action space, which may be detrimental in settings like fast locomotion (Appendix H), but this may be fixed by more clusters or a residual correction (Shafiullah et al., 2022). In addition, like prior work execution of our subwords is open loop, so exploration can be inefficient (Amin et al., 2020) and unsafe (Park et al., 2021). Finally, in order to operate on the CoinRun domain, we downsample inputs from $64 \times 64$ resolution to $32 \times 32$ to make matrix inversion during merging less expensive (2 hours vs. 2 minutes). In high-dimensional visual input domains, our merging may be too computationally expensive to perform. However, this can be resolved by using neural network features instead of images. We also speculate that higher-quality demonstrations could allow us to generate skills simply by merging based on frequency (Table 1, CoinRun), and these demonstrations may be easy to obtain if they don't need to be collected in the deployment domain (Table 5).

Architectures from NLP have made their way into offline RL (Chen et al., 2021; Janner et al., 2021; Shafiullah et al., 2022), but as we have demonstrated, there is a trove of further techniques to explore. Given prior evidence, and the experiments in Appendix C, that discretization can be helpful in offline RL, we leveraged such discretization to form skills through a simple tokenization method. Such a method is much faster both in skill generation and in policy inference, and leads to strong performance in a relatively small sample budget on several challenging sparse-reward tasks. Moreover, the discrete nature of our skills lends itself to interpretation: one can simply look at the execution to figure out what has been extracted (Appendix B). Given its many advantages, we believe that such a tokenization method is the first step on a new road to efficient reinforcement learning.

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
