# A    ABLATIONS

Certainly the level of discretization, and the size of the vocabulary will have an effect on performance. In the following sections we perform ablations over the primary hyperparameters on AntMaze-Medium and Kitchen.

## A.1    NUMBER OF DISCRETE PRIMITIVES

All of our results in Table 1 use the simple rule-of-thumb that $k = 2 \times$ degrees-of-freedom. Such a choice may not be optimal depending on the setting. In Table 6 we see that this choice seems to be a simple sweet spot across the two domains, though the method can achieve reward with significantly different values of $k$.

Table 6: Results for different numbers of clusters in terms of the number of degrees-of-freedom (DoF). AntMaze DoF = 8, Kitchen DoF = 9. The default setting is in bold.

| $k$ | 4 | $1 \times \text{DoF}$ | $\mathbf{2 \times DoF}$ | $4 \times \text{DoF}$ | $8 \times \text{DoF}$ |
|---|---|---|---|---|---|
| antmaze-medium-diverse | 0.0 | 0.0 | $0.40 \pm 0.55$ | $0.20 \pm 0.45$ | 0.0 |
| kitchen-mixed | $0.16 \pm 0.35$ | $0.08 \pm 0.18$ | $0.72 \pm 0.40$ | 0.0 | $0.20 \pm 0.45$ |

## A.2    MAXIMUM VOCABULARY SIZE

A crucial property of the vocabulary is the length of the subwords within. Long subwords lead to more temporal abstraction and easier credit-assignment for the policy, but long subwords can also get stuck for many transitions, leading to poor exploration. In Table 7, we vary the value of $N_{\max}$, which is a proxy for the length of the subwords in the vocabulary. Our default setting for each environment targets an average length of around 10 to match the baselines, but we see that different domains may have different optimal choices for length, which makes sense given the episode length for Kitchen is around a quarter of that of AntMaze.

Table 7: Results for maximum vocabulary size (proxy for length).

| $N_{\max}$ | 32 | 64 | 128 | 256 | 512 |
|---|---|---|---|---|---|
| antmaze-medium-diverse | 0.0 | $0.25 \pm 0.5$ | $0.40 \pm 0.55$ | $0.61 \pm 0.48$ | $0.07 \pm 0.08$ |
| kitchen-mixed | 0.0 | $0.50 \pm 0.57$ | $0.72 \pm 0.40$ | $0.04 \pm 0.10$ | 0.0 |

## A.3    MINIMUM VOCABULARY SIZE

Ultimately, the dimensionality of the action space will make exploration easier or harder. A large vocabulary results in too many paths for the policy to explore well, but a vocabulary that is too small may not include all the subwords necessary to represent a good policy for the task. We see in Table 8 that even if AntMaze can be accomplished with fewer subwords (a smart handcrafted action space might consist of one action for turning and one for moving forward), Kitchen performance suffers significantly at low values.

Table 8: Results for minimum vocabulary size $N_{\min}$. In bold is the default setting.

| $N_{\min}$ | 4 | 6 | 8 | 12 | $\mathbf{16}$ |
|---|---|---|---|---|---|
| antmaze-medium-diverse | 0.0 | $0.24 \pm 0.49$ | $0.71 \pm 0.41$ | $0.41 \pm 0.49$ | $0.40 \pm 0.55$ |
| kitchen-mixed | 0.0 | 0.0 | 0.0 | $0.01 \pm 0.01$ | $0.72 \pm 0.40$ |

# B   QUALITATIVE DESCRIPTION OF SKILLS

One nice property of our method is that, given that we create a finite and discrete vocabulary, we can inspect the discovered skills. Below, we discuss the AntMaze and Kitchen domains as an example. In order to visualize skills, we take the subwords and execute them for 200 steps in the environment, and visualize the resulting trajectory. It may be the case that the actual duration of a skill could be much shorter, but this is done to make the motions very clear.

In Figure 3 (main paper), we see all the skills extracted for a run of AntMaze. In particular, turning in both directions, with differing turn radii, as well as linear motions in different directions are discovered. It is straightforward to imagine why one would need both in designing an action space, and it seems that there are few explicit repetitions (though many variations on a theme) in the discovered skills. Also, as desired, the skills accomplish some coherent motion, instead of just repeating the same action and staying in place as a result, or falling over due to an unstable execution, or jittering randomly.

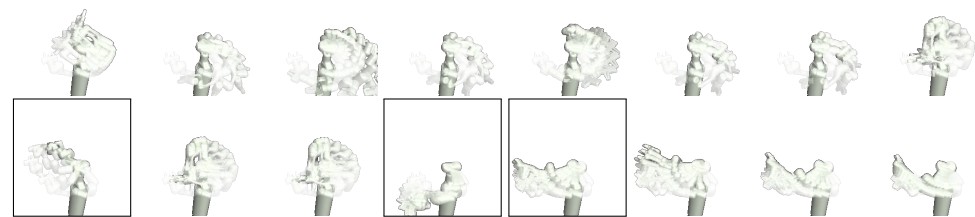

Figure 6: All skills generated for `kitchen-mixed-v0` where the transparency is higher for poses earlier in the trajectory. We see a range of different behaviors across the skills. Framed skills are highlighted in more detail in the text.

In Figure 6, we visualize the different skills discovered in the Kitchen domain. These are difficult to present in a static form, as it is not simple to visualize interaction with the environment, but they consist of a variety of reaching and rotational motions that are useful for interacting with different objects. In the bordered images, we highlight three particular skills. In the bottom left is a reaching skill that might be used for reaching the light switch/oven knobs. Next from the left is a turning skill that could be useful for adjusting some knob if the arm is in a particular position. Lastly there is a pulling skill, that might be useful for opening the microwave door. In general, these skills may not make sense unless the arm is already in a particular starting position, which makes visualizing them nontrivial.

# C   DISCRETE ACTIONS FROM DATA

Prior work combines discretization with many additional architectural and optimization components. To test the behavior of discrete actions in isolation, we perform behavior cloning with a simple fully-connected neural network on demonstration data from the D4RL (Fu et al., 2020) dataset. To be clear, our objective is not to show that simple $k$-means on demonstrations outperforms contemporary methods. Instead, we investigate whether behavioral cloning with these actions achieves modest performance in which case there is the potential for further tokenization to be effective in sparse-reward domains.

We compare to CQL (Kumar et al., 2020), an offline Q-learning algorithm that encourages staying close to the demonstration distribution; Diffuser (Janner et al., 2022), a diffusion model conditioned on an initial and final state; and Diffusion-QL (Wang et al., 2022), an offline Q-learning algorithm that uses a diffusion model on top of actions to stay close to the demonstration distribution. For more details on the experimental setting, see Appendix D.

In Table 9, we see that the dense-reward locomotion domains suffer from discretization, which makes sense as locomotion policies may require fine-grained control to move at high speed and achieve high reward. On AntMaze, however, we see that simple $k$-means discretization significantly boosts performance. This can be due to the fact that, at a given position, there are many possible motions

Table 9: D4RL offline learning results. BC numbers are from Emmons et al. (2021), Diffusion-QL numbers are from Wang et al. (2022), CQL numbers are from Kumar et al. (2020). $k$-means BC numbers are from the checkpoint with the best average score during training.

| Task | BC | $k$-means BC | CQL | Diffuser | Diffusion-QL | $k$-means BC + goals |
|---|---|---|---|---|---|---|
| hopper-medium | 52.9 | $8.3_{\pm 1.9}$ | 58.0 | $74.3_{\pm 1.4}$ | $90.5_{\pm 4.6}$ | — |
| hopper-medium-replay | 18.1 | $8.3_{\pm 1.6}$ | — | $93.6_{\pm 0.4}$ | $101.3_{\pm 0.6}$ | — |
| hopper-medium-expert | 52.5 | $10.2_{\pm 1.7}$ | 111.0 | $103.3_{\pm 1.3}$ | $111.1_{\pm 1.3}$ | — |
| walker2d-medium | 75.3 | $9.8_{\pm 2.7}$ | 79.2 | $79.6_{\pm 0.55}$ | $87.0_{\pm 0.9}$ | — |
| walker2d-medium-replay | 26.0 | $7.9_{\pm 0.7}$ | 0 | $70.6_{\pm 1.6}$ | $95.5_{\pm 1.5}$ | — |
| walker2d-medium-expert | 107.5 | $9.7_{\pm 0.6}$ | 98.7 | $106.9_{\pm 0.2}$ | $110.1_{\pm 0.3}$ | — |
| halfcheetah-medium | 42.6 | $27.2_{\pm 3.5}$ | — | $42.8_{\pm 0.3}$ | $51.1_{\pm 0.5}$ | — |
| halfcheetah-medium-replay | 36.6 | $8.6_{\pm 2.8}$ | — | $37.7_{\pm 0.5}$ | $47.8_{\pm 0.3}$ | — |
| halfcheetah-medium-expert | 55.2 | $15.1_{\pm 4.4}$ | 62.4 | $88.9_{\pm 0.3}$ | $96.8_{\pm 0.3}$ | — |
| antmaze-umaze | 54.6 | $84.0_{\pm 8.3}$ | 74.0 | — | $93.4_{\pm 3.4}$ | $82.6_{\pm 6.6}$ |
| antmaze-umaze-diverse | 45.6 | $93.8_{\pm 4.7}$ | 84.0 | — | $66.2_{\pm 8.6}$ | $89.0_{\pm 7.2}$ |
| antmaze-medium-play | 0.0 | 0.0 | 61.2 | — | $76.6_{\pm 10.8}$ | $15.2_{\pm 9.8}$ |
| antmaze-medium-diverse | 0.0 | 0.0 | 53.7 | — | $78.6_{\pm 10.3}$ | $14.4_{\pm 7.5}$ |
| antmaze-large-play | 0.0 | 0.0 | 15.8 | — | $46.4_{\pm 8.3}$ | $2.6_{\pm 2.8}$ |
| antmaze-large-diverse | 0.0 | 0.0 | 14.9 | — | $56.6_{\pm 7.6}$ | $10.8_{\pm 5.6}$ |
| kitchen-complete | 65.0 | $54.0_{\pm 3.5}$ | 43.8 | — | $84.0_{\pm 7.4}$ | — |
| kitchen-partial | 38.0 | $14.8_{\pm 0.2}$ | 49.8 | — | $60.5_{\pm 6.9}$ | — |
| kitchen-mixed | 51.5 | $18.0_{\pm 4.6}$ | — | — | $62.6_{\pm 5.1}$ | — |

that can move the body, but they are completely distinct in action space, which a unimodal policy may fail to capture. In the Kitchen domain, a policy that reasons over discrete actions achieves modest performance. The data in this domain was collected from expert human demonstrations, and there is very low variability in the executions, so it may be the case that multimodality is simply not necessary.

## D  OFFLINE RL EXPERIMENTAL DETAILS

### D.1  DATA

To measure the performance of behavior cloning with discrete actions, we take advantage of datasets from D4RL (Fu et al., 2020). In particular, we select three subclasses of tasks. First are the MuJoCo dense-reward locomotion tasks, which consist of demonstrations collected from an RL agent, where `medium` refers to a policy partway through optimization, `medium-replay` refers to all samples in the replay buffer til the policy obtains `medium` performance, and `medium-expert` refers to a mix of data from an expert policy and a policy midway through training. Second are the AntMaze tasks, which are a collection of sparse-reward maze navigation tasks on top of the MuJoCo Ant quadrupedal robot. Demonstrations are either `play`, which is a scripted policy navigating between a couple fixed start and endpoints, which do not overlap with the final task, and `diverse`, which is the same scripted policy navigating between random start and endpoints. Third, are the Kitchen tasks, which are a collection of VR-collected demonstrations of subtasks in a Kitchen, where the final goal is to perform 4 in sequence. The settings include `complete` that consists of demonstrations of all subtasks in order, `partial` that consists of some sequences in the correct order, and others not, and `mixed` that consists of subtask demonstrations, some of which are unused for the final task.

To perform the goal-conditioned experiments for the AntMaze task, for each trajectory we extract the last state that is considered "terminal" (e.g., falling over or reaching the goal) and create a (state, action, goal) triplet for each transition in the trajectory.

### D.2  MODEL

For the model, we choose a 4-layer MLP with 256 hidden units in each layer. We use the default initialization in Stable Baselines 3 (Raffin et al., 2021).

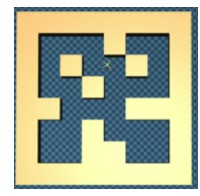 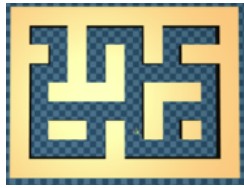 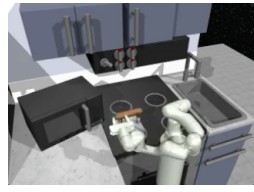 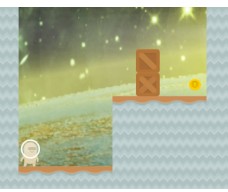

(a) `antmaze-medium`    (b) `antmaze-large`    (c) `kitchen`    (d) `CoinRun`

Figure 7: Offline environments, figures courtesy of Fu et al. (2020) and Cobbe et al. (2019). For mazes, the starting locations are in the bottom left, and goals are in the top right.

## D.3 OPTIMIZATION

We train our model with Adam (Kingma and Ba, 2014) with a learning rate of $3e - 4$ and the default PyTorch (Paszke et al., 2019) betas. All numbers are reported for 5 random seeds with 300 epochs of training.

## D.4 CHOICE OF NUMBER OF CLUSTERS

For the locomotion and AntMaze environments, we choose $k = 2\times$ the number of DoF. For Kitchen, we choose $k = 8\times$ DoF. This discrepancy is due to the fact that Kitchen performs poorly with the simple baseline choice. In particular we believe that this is due to the fact that Kitchen demonstrations are particularly good, and not particularly multimodal, so they benefit from the higher resolution that a larger $k$ affords. This is similar to the hyperparameter settings of Shafiullah et al. (2022) in the same environment.

## D.5 IMPLEMENTATION

Code was implemented in Python using PyTorch (Paszke et al., 2019) and PyTorch Lightning (Falcon) for deep learning, and Weights & Biases (Biewald, 2020) for logging.

## D.6 COMPUTATIONAL REQUIREMENTS

All experiments were performed on an internal cluster with access to around 100 Nvidia 2080 Ti (or more capable) GPUs. Each single run fits in around 2 GB of GPU memory on a single machine. For supervised learning, training takes less than 2 hours on a single machine.

# E ONLINE RL EXPERIMENTAL DETAILS

## E.1 DATA

As a set of diverse and challenging sparse-reward tasks, we select AntMaze and Kitchen from D4RL (Fu et al., 2020) and CoinRun (Cobbe et al., 2019).

AntMaze (Figs. 7(a) and 7(b)) is a task where a MuJoCo Ant robot is tasked with solving a maze. The observation space consists of positions and joint angles of the body geometries, while actions correspond to joint torques. Crucially, no information about the maze layout is given, so the agent must learn this through exploration. Reward is 0 unless within a small distance $\epsilon$ of the goal, in which case it is 1. Demonstrations from the dataset consist of a non-RL agent navigating between random start and end points within the maze. In particular, the demonstrations are highly suboptimal, often crashing into walls, flipping over, and getting stuck.

Kitchen (Fig. 7(c)) is a task where a Franka Panda arm is tasked with performing a set sequence of 4 subtasks in a mock kitchen environment. Example subtasks might be moving a kettle between burners, turning on the stove, or opening the microwave. Observations consist of position and joint angles of the arm, as well as positions of key objects to be manipulated, and actions are joint torques. Once again, no information about the layout is given to the agent and must be learned through exploration.

Rewards are $0$ unless the correct subtask is completed in the correct order, which yields a reward of $1$. The $4$ subtasks must be completed, so there is a maximum reward of $4$ available. Demonstrations are collected by humans using a VR interface, and consist of near-perfect executions of different sequences of 4 subtasks from the final subtask sequence.

CoinRun (Fig. 7(d)) is a procedurally-generated platforming game intended to mimic classic games that involves traversing obstacles and avoiding enemies in order to reach a final goal. Each level has a different layout and visual style, designed by humans, in order to require more general recognition from the policy. Observations consist of a $64 \times 64$ visual observation of the scene, centered on the agent, with velocity information painted into the upper-left corner. Actions are discrete and consist of moving, jumping, and staying still. Reward is $0$ until the final goal for a level is reached, in which case it is $10$. For RL, we select a fixed subset of 10 "hard" levels in sequence for an agent to complete, to mimic classic games, so the maximum possible reward is $100$. Demonstration data is collected by us through playing around $100$ "easy" levels with different layout and visual style than the eventual levels we perform RL on.

### E.2 MODEL

For the model, we choose a 4-layer MLP with 256 hidden units in each layer. We use the default initialization in Stable Baselines 3 (Raffin et al., 2021).

### E.3 OPTIMIZATION

For our RL agent, we use SAC-discrete (Christodoulou, 2019). Both critics as well as the policy are optimized with Adam with a learning rate of $3e-4$. Replay buffer size is set to the standard 1 million transitions. We update both critics and the policy every step of environment interaction and sample uniformly from the replay buffer to do so. Unlike Christodoulou (2019), we follow a similar convention to Haarnoja et al. (2018b) and automatically optimize $\alpha$. We choose a target entropy of $-\log|\mathcal{V}|$, except for CoinRun domains, where we use $\frac{1}{2} \cdot \log|\mathcal{V}|$. A negative target entropy may not make sense for a discrete distribution, but we found that any other choice led to extremely unstable optimization due to runaway $\mathcal{Q}$ estimates. This hints that SAC may not be well-adjusted to discrete-action sparse-reward domains, as argued by Zhou et al. (2022).

For AntMaze we train for 10 million steps, for Kitchen we train for 2 million, and for CoinRun we train for $500,000$ or til policy divergence. All numbers come from 5 random seeds of training, evaluated over 100 rollouts of the deterministic policy. To avoid biasing numbers, we simply report the final average deterministic performance of the policy, even in cases when performance is better earlier in training.

### E.4 SKILL-EXTRACTION HYPERPARAMETERS

For AntMazes we choose defaults of $k = 2 \cdot d_{\text{act}}$, $N_{\max} = 128$ and $N_{\min} = 16$. For Kitchen we choose defaults of $k = 2 \cdot d_{\text{act}}$, $N_{\max} = 256$ and $N_{\min} = 16$. For CoinRun there is no need for discretization, so we only choose $N_{\max} = 64$ and $N_{\min} = 16$. These defaults are chosen to approximately match the length 10 skills of SSP, as the choice of $k$ and $N_{\max}$ will directly influence the length of discovered skills.

### E.5 IMPLEMENTATION

Code was implemented in Python using PyTorch (Paszke et al., 2019) for deep learning, Stable Baselines 3 (Raffin et al., 2021) for RL, and Weights & Biases (Biewald, 2020) for logging.

### E.6 COMPUTATIONAL REQUIREMENTS

All experiments were performed on an internal cluster with access to around 100 Nvidia 2080 Ti (or more capable) GPUs. Each single run fits in around $2\,\text{GB}$ of GPU memory on a single machine. On AntMaze, training for our method typically takes around 10 hours for a single run, while SSP (Pertsch et al., 2021) takes 12 hours and SFP (Bagatella et al., 2022) takes over a week. In particular, this

highlights exactly how poor the scaling can be for methods that call a large model at every transition. More precise information is available in Table 3 (main paper).

## F    NOTES ON REPRODUCIBILITY

One important note to draw from this work is that the results are not always stable. Such inconsistency goes beyond our work alone: the disagreement of dense-reward offline RL (Fu et al., 2020; Emmons et al., 2021; Janner et al., 2021; Wang et al., 2022) numbers; the failure to reproduce SSP baseline results Table 1; and large standard deviations and unclear trends across Tables 1, 6, 7, and 8. In our case, there are a few sources of nondeterminism. We use Scikit-learn (Pedregosa et al., 2011) for our $k$-means implementation, which yields slightly different results depending on the CPU even with the same seed, which then leads to slightly different skills in the downstream merging process (though they are largely of the same categories). In addition, the library we use for RL, Stable Baselines 3 (Raffin et al., 2021), has subroutines that cannot be controlled on the GPU. Finally, we often observe collapse of the policy during training, which is not an unfamiliar issue in RL. This could be due to the design of SAC (Haarnoja et al., 2018b), which may not easily adapt to the discrete or sparse-reward setting (Zhou et al., 2022), leading to further instability. All the above suggests that five random seeds is not enough to quantify performance (Henderson et al., 2018), however running more samples incurs a significant computational burden, which is not a substantial issue for our method, but is particularly burdensome for baselines. Still, results on exploration in Section 4.2 give us confidence that our modification is working as desired, and we hope that a method like ours may lead to stronger and faster sparse-reward RL in the future.

## G    DATA QUANTITY

To see how our method performs under limited quantities of data, we subsample the trajectory datasets before generating subwords. We see in Table 10 that less data does not always correlate with worse performance, though the results are mixed as to what is the best setting. Such a result is due to the fact that our subword extraction method only merges the skill that moves "farthest," thus the amount of distracting data is not a core issue, but rather the existence of good skills within that data.

Table 10: Experiments across domains for our method when data is subsampled, by percentage of the original dataset. We see that performance is rather uncorrelated with dataset percentage, which is a result of our subword extraction pipeline.

| Task | 10% | 25% | 50% | 100% |
|---|---|---|---|---|
| antmaze-medium-diverse | $0.99_{\pm 0.02}$ | $0.20_{\pm 0.45}$ | $0.80_{\pm 0.45}$ | $0.40_{\pm 0.55}$ |
| antmaze-large-diverse | 0.0 | 0.0 | 0.0 | $0.34_{\pm 0.46}$ |
| kitchen-mixed | $0.20_{\pm 0.04}$ | 0.0 | $0.20_{\pm 0.04}$ | $0.72_{\pm 0.40}$ |
| CoinRun | $3.44_{\pm 1.40}$ | $3.40_{\pm 1.73}$ | $2.72_{\pm 2.22}$ | $2.90_{\pm 2.90}$ |

## H    EFFECT OF DISCRETIZATION IN LOCOMOTION

As mentioned in our Limitations, discretization may remove resolution from the action space that could be useful, in particular for fine-grained manipulation or fast-locomotion tasks. To study this limitation, we investigate the effect of varying the discretization level on the Hopper locomotion task from D4RL (Fu et al., 2020). We use SaS-freq with $N_{\min} = 32$, $N_{\max} = 128$, training 5 seeds for 3 million steps each.

In Table 11, we see that the conclusions are quite straightforward. More discretization hurts performance, where higher $k$ recovers more of the original action space as smaller regions are clustered together. For simplicity in our sparse-reward results, we used a relatively small number of clusters ($2 \cdot d_{\text{act}}$), but there is no reason why a larger number could not be used in domains that require it.

Table 11: Experiments on the `hopper-expert` domain for varying number of cluster $k$. Coarser discretization is worse.

| $k$ | SaS-freq Reward |
|---|---|
| 12 | $2813.3_{\pm 104.4}$ |
| 32 | $3182.6_{\pm 335.4}$ |
| 64 | $3248.8_{\pm 137.8}$ |

## I  EFFECT OF DATA QUALITY

To see how our method performs under different kinds of data quality, we study SaS-freq on the Hopper task from D4RL (Fu et al., 2020). This is because, unlike sparse-reward tasks considered in the rest of the paper, Hopper provides a clear delineation of demonstration quality: `random` for transitions from a random policy, `medium` for transitions from a policy partway through training, and `expert` for transitions from a policy at the end of training. We set $k = 12, N_{\min} = 32, N_{\max} = 128$ and train 5 seeds for 3 million steps.

From Table 12, expert demonstrations provide the best skills, but surprisingly random demonstrations are much more competitive than skills from a medium policy. On further inspection, this is because medium demonstrations contain long segments of the policy standing, which it learns before walking quickly, so the skills discovered primarily relate to standing. In the case of random demonstrations, the skills are quite short in length, so through RL the policy can learn to recombine them. For expert demonstrations this is similar, but the skills are of higher quality.

Table 12: Experiments on the Hopper domain for varying data quality. Random demonstrations outperform medium demonstrations as the skills extracted are much shorter for equivalent hyperparameters, so during RL the policy learns to recombine them.

| Task | SaS-freq Reward |
|---|---|
| `hopper-random` | $2607.3_{\pm 122.0}$ |
| `hopper-medium` | $980.2_{\pm 2.0}$ |
| `hopper-expert` | $2916.1_{\pm 129.3}$ |

## J  Q-VALUE COLLAPSE

Here we provide visualizations of the Q-function during RL for one seed of SaS-dist on `antmaze-umaze` that shows good exploration, but at the end of training achieves no reward. We see in Figure 8 that initially Q-estimates are highest on the frontier, but as training progresses, Q-estimates equalize and drive the policy to uniform behavior, which eventually ruins exploration. Combatting such collapse is a large priority in the future for making exploration much more stable.

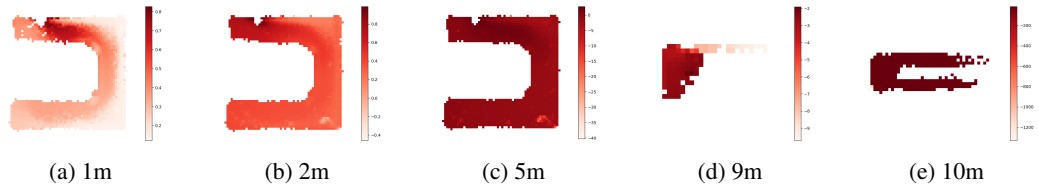

(a) 1m      (b) 2m      (c) 5m      (d) 9m      (e) 10m

Figure 8: Q-value for locations in the replay buffer during RL for a seed where optimization collapses. Initially Q-values are aligned with the task, but as optimization progresses, Q-values equalize, which leads to collapse to a random policy.