# OpenReview forum: "Subwords as Skills: Tokenization for Sparse-Reward Reinforcement Learning"
_ICLR.cc/2024/Conference — Submitted to ICLR 2024_

### Official Review · Reviewer_Y696 · 2023-10-21

**Soundness:** 2 fair
**Presentation:** 3 good
**Contribution:** 3 good
**Rating:** 6
**Confidence:** 3

**Summary:**

The authors propose a method to build a set of skills from a set of demonstrations by discretizing the action space, identifying elementary tokens and binding these tokens together into skills. The corresponding skills are not conditioned on an input state. The authors show that their method is faster to learn and more efficient than relevant baselines in a variety of ant-maze demonstrations, Kitchen demonstrations and CoinRun.

After a first of changes performed by the authors, I moved my score from 5 to 6.

**Strengths:**

- The authors made a set of clever design choices to get an efficient method: learning from demonstrations, discretizing actions, learning skills without training a neural network, avoiding conditioning on states or observations.

- They performed a good bibliographic study and identified relevant baselines

- The experimental study addresses good questions

**Weaknesses:**

- Several points are unclear (see questions below)

- Some important results which do not speak in favor of the method are deferred to the appendices

- The related work in the main paper could be more focused (see below)

If the authors manage to produce a next version that properly answers my questions below, I will be happy to switch my evaluation towards acceptance.

**Questions:**

### Questions

- In Table 1, are the antmaze results obtained by conditioning on the state or on images observations? I suspect this is the state. If I'm right, how did you adapt SSP and SFP, which are designed to work with images? Isn't this comparison unfair, since SSP and SFP, which are designed to work with images?
- Still in Table 1, I was wondering why you did not compare to SpiRL and SpiRL-cl, which are defined in the same paper as SSP (SSP is a sort of ablation of SpiRL). After reading Appendix G, it happens that you did so, but the results speak less in favor of your method. This is somewhat unfair.
- From the above 2 questions and remarks, my feeling is that to be fair, the main paper should have two tables like Table 1, one with learning from states and one with learning from images, where the separation of baselines and the environments all make sense. Eventually, if you don't have enough room, I think you could move all SSP results to an Appendix, as SSP is just an ablation of SPiRL. The fact that your method does not establish a new SOTA on all these results does not matter much, it has interesting enough features to deserve publication if properly presented.
- In Table 3, SFP looks faster than SSP, whereas the paper seems to mention the contrary in several places. Can you clarify this?
- Is Table 3 obtained when learning from images or from states?
In Table 1 (and/or 3 and/or 9), it would ne nice to specifiy the number of obtained skills with your methods and the baselines. If this is a hyper-parameter, this should be made clear too.
- I did not find a list of hyper-parameters of your method. BTW, it would be nice to give a name to your method.

- Section 4.2 could be made clearer. At first read, it is unclear whether you are (1) using RL "from scratch", (2) you are learning from a dataset of demonstrations or (3) something intermediate. My guess is that up to the first half of the second paragraph, you speak about (1). But then, after "Moreover", you speak about weaknesses of SFP and SSP in the context of (2), without proper articulation of the ideas. And the remark about SAC goes back to (1). Again, Figure 5 displays "state visitation" of your method, but if your method is learning from a dataset, this makes no sense. All this must be clarified, including the title of the section.
- Note also at the end of the paragraph: "the large error bars of our method are a result of an optimization failure", we don't have pictures with error bars in the paper, just std info in some tables.

### Local issues

- Figs 1 and 3 are not much informative, particularly Fig. 1b is not readable

- The first paragraph of Section 4.1 is more about related work than about experiment. I think you should have a more focused related work section (mostly the "learning skills from demonstration" part) where you can give more details about these methods and reject what is less relevant into an Appendix. Another approach is to have a "Baselines" section where you describe the baselines with enough details so as to make the point you made in this paragraph.

- I appreciate the honesty of Appendix F, but it cast doubts on the validity of your results. As advocated in [1], if you don't have the compute to get statistically valid results, work on cheaper tasks :)
[1] Patterson, A., Neumann, S., White, M., & White, A. (2023). Empirical Design in Reinforcement Learning. arXiv preprint arXiv:2304.01315.

- p9: results in Table 9 -> you must mention it is in an Appendix (but you should rather reorganize, see above).


### Typos

- p4: euclidean -> Euclidean
- p8: brownian -> Brownian

---

> ### Author Response · Authors · 2023-11-15
> **Thanks for your review!**
>
> Thank you for spending time with our paper! We aspire to answer your comments in detail below, and believe they have significantly improved the paper! We use purple for revisions related to your comments.
>
> ### Strengths
>
> Thanks for acknowledging this. We hope such efficiency leads to much faster RL in the future.
>
> ### Weaknesses
>
> We will address the questions in detail below, thank you for the encouragement.
>
> ### Questions
>
> > ...SSP and SFP, which are designed to work with images?...
>
> You're correct that we use state-based observations. As far as we understand, SSP and SFP are agnostic to the choice of state or image-based domains as they only model action sequences (SSP is a VAE, and SFP is a sequence model). SSP has experiments in the state-based Kitchen (for which we use their official code), and SFP's primary experiments are in meta-world which offers state and visual variants. Figure 7 in the SFP paper is for state-based, while Figure 8 is for visual. Both of these methods learn skills regardless of the observation space, and they assume the same information of our method.
>
> > ...why you did not compare to SPiRL and SPiRL-cl...
>
> We have made changes to Sections 4.1 and 4.3 to address this. The reason that we see the correct baseline as SSP and not SPiRL is that we extract skills without any information on when to use those skills, then learn a policy through RL which must find the association. SSP does the exact same with a VAE. SPiRL extracts both skills, and knowledge of when to use those skills in the environment, biasing the RL exploration. This leads to faster and more stable learning when the final task aligns with the demonstrations, but handicaps exploration when in a new environment. If we wanted to compare with observation-conditioned methods correctly, we should design a prior that connects our subwords to the observations they are used in, and biases exploration in RL like SPiRL. As we did not do this, it seems correct to us to compare directly against SSP.
>
> > ...the main paper should have two tables like Table 1...
>
> Thank you for the encouraging words! We believe that there is a misunderstanding due to our poor presentation, and we hope the above clarifies this. We've changed Sections 4.1 and 4.3 to make these points more clear, including moving Appendix G into Section 4.3. Our original goal was to improve exploration, even in settings where demonstrations are not tightly aligned, so performance against methods with more assumptions was not the primary point.
>
> > ...SFP looks faster than SSP...
>
> SFP is faster to train, but the online rollout is 4x slower. During RL, the rollout speed is what makes SFP painfully slow.
>
> > Is Table 3 obtained...
>
> Thanks for these important clarifications. We used the same settings as our primary experiments in antmaze-medium-diverse, that is 16 skills with average length in Table 2 attempting to be consistent with length-10 skills of SSP. We detail most hyperparameters in Appendix E and reference this in the text. It appears we also missed the inclusion of default settings for N_min and N_max, which we add now in 4.1.
>
> > I did not find a list of hyper-parameters...'
>
> Thanks for the catch! Please see our answer above on hyperparameters. We've also added a simple abbreviation (SaS for subwords as skills) to Section 3, which is used in all tables and figures.
>
> > Section 4.2 could be made clearer...
>
> This is very helpful feedback. We make changes to the abstract at the introduction to clarify. We also added small paragraphs at the beginning of Section 3 and Section 4. We also retitle Section 4.1 and 4.3 to more directly address the problem setting.
>
> Our method is
> 1. extract actions from reward-free demonstrations without any knowledge of when to use them'
> 2. do RL from scratch on a task with the modified action space
>
> Hopefully these changes further clarify!
>
> > Note also...
>
> Thanks for the catch, we've edited the wording to "standard deviations."
>
> > Local Issues
>
> > Figs 1 and 3...
>
> We found it important to provide some visualization, though it's quite difficult to do so for Kitchen. We discuss this more in Appendix B, which we now reference in the caption.
>
> > The first paragraph...
>
> Thanks for this comment, we've now condensed this discussion in Section 4.1 and the related work, and grouped the baselines paragraph as suggested.
>
> > ...honest of Appendix F...
>
> Thanks for this comment. We have suffered ourselves trying to reproduce SSP and SPiRL, which is why it is included. One issue is that cheaper tasks (like 2d pointmazes used in SSP and SFP) are easy enough that all methods achieve perfect performance. We also attempt to work on domains inherited from other papers (OPAL, SPiRL). It's tricky to find a faster task to simulate that's nontrivial...
>
> > ...results in Table 9...
>
> We've reorganized as mentioned.
>
> > Typos
>
> Nice catches, we've made changes.
>
> Thanks for your detailed read and many comments. All of your feedback is very helpful in improving the paper!

---

> > ### Comment · Reviewer_Y696 · 2023-11-16
> > **Good job**
> >
> > I thank the authors for their efficient treatment of my remarks and, as far as I can see, of some of the other reviewers'. Some of the answers have improved my opinion about this paper, I'll increase my score to "marginally above acceptance threshold".
> >
> > Two points after looking at the paper again:
> > - I find it ugly to have the first figure on top of the abstract. That's a matter of taste, I'll understand if the authors do not want to change this (maybe they should show me that they are not the only ones to do so)
> > - in the paper edits, the \cite and \citep commands have not been appropriately managed. The authors should double check this, I spotted several places where a \cite should be turned into a \citep, mostly in the introduction and related work.

---

> > > ### Author Response · Authors · 2023-11-16
> > > **Thanks for your quick response**
> > >
> > > We appreciate you taking the time to look over the rebuttal, and are glad it has convinced you to change your score.
> > >
> > > On the two additional points:
> > > > I find it ugly...
> > >
> > > Here's a small list of ICLR 2023 papers with the figure first. The convention of figure first is also pretty common in computer vision conferences. We prefer this arrangement, though we agree it's a matter of taste and we're not wed to it.
> > > - Hong et al. 2023. https://openreview.net/pdf?id=g7U9jD_2CUr
> > > - Asano and Saeed 2023. https://openreview.net/pdf?id=6kxApT2r2i
> > > - Ma et al. 2023. https://openreview.net/pdf?id=IDJx97BC38
> > > - Zhang et al. 2023. https://openreview.net/pdf?id=W918Ora75q
> > > - Liu et al. 2023. https://openreview.net/pdf?id=0cpM2ApF9p6
> > > - Huang et al. 2023. https://openreview.net/pdf?id=W-nZDQyuy8D
> > > - Kwon and Ye 2023. https://openreview.net/pdf?id=Nayau9fwXU
> > > - Sheynin et al. 2023. https://openreview.net/pdf?id=x5mtJD2ovc
> > >
> > > > ...the \cite and \citep commands have not been appropriately managed...
> > >
> > > Thanks for the catch, they should be fixed now.

---

> > > > ### Comment · Reviewer_Y696 · 2023-11-19
> > > > **OK, your list is convincing, forget my comment :)**
> > > >
> > > > Everything is in the title :)

---

> > > > > ### Author Response · Authors · 2023-11-21
> > > > > **Does it tilt the scale any further? :)**
> > > > >
> > > > > See title, thanks for all your engagement.

---

### Official Review · Reviewer_NHza · 2023-10-30

**Soundness:** 1 poor
**Presentation:** 2 fair
**Contribution:** 2 fair
**Rating:** 3
**Confidence:** 4

**Summary:**

This paper proposes to extract skills from offline dataset via tokenization, providing temporal abstractions for downstream RL. The continuous actions in the dataset are clustered into discrete tokens. Then, similar to the tokenization in NLP, consecutive tokens are merged into new tokens recursively and some tokens are pruned. The paper proposes a metric based on change of the observation caused by the action token to select tokens to merge and prune. Experiments in AntMaze, Kitchen and CoinRun shows that the method outperforms some state-free skill learning baselines and vanilla SAC. The method also has advantages in computation efficiency, exploration ability, and domain generalization compared with baselines.

**Strengths:**

1. This paper brings a novel idea that uses the technique of tokenization in NLP to provide temporal abstract actions for RL.

2. The experiments evaluate various aspects of the proposed method, including performance, computation efficiency, exploration behavior, and domain generalization, which are all important for skill learning methods.

**Weaknesses:**

1. About paper writing: In Introduction, the background of RL and exploration is too long, while the study only  focuses on the topic of skill discovery from data, which has been extensively studied. In Related Work, the section of "Skill Learning from Interaction" is also unnecessary because it is in parallel with your topic.

2. About the observation distance-based score: Since it is complicated and lacks a theoretical explanation, more intuitive explanations or visualization is needed to illustrate this formula. In many domains with image observations, partial observations or complicated robotic tasks, the Euclidean distance in the observation space may not work.

3. Soundness of experiments:
- Lack a state-based baseline in Table 1 experiments, like SPiRL. SPiRL reaches a return of 2.9 [1] in Kitchen, which outperforms the proposed method a lot. It may also perform well in other environments.
- Training curves of returns are usually required for RL experiments, which depict both performance and sample efficiency. However, this paper only presents the return at the end of training in Table 1, which can be doubtful due to the instability of RL algorithms. The data in Table 1 have large variance and are therefore unreliable.

4. Lack the discussion of disadvantages of state-free skills: The paper emphasizes the generalization ability of state-free skills, but ignores its disadvanteges. State-free methods are inefficient on large datasets with diverse behaviors, since a large skill space is required to model all the behaviors. In this case, state-conditioned methods like SPiRL can model possible behaviors based on the state, thus providing a compact skill space. The paper also lacks such study on the influence of data diversity.

[1] Accelerating Reinforcement Learning with Learned Skill Priors

**Questions:**

1. I suggest that the author revise the paper to introduce the topic of skill discovery more directly and give a problem formulation before the section of Method.

2. In action space clustering, the number of clusters is very small ( $2\times d_{act}$ is less than 20 in AntMaze and Kitchen). Is this because the agent behavior in the dataset is not diverse?
If the dataset is large or collected by various stochastic policies, a small number of clusters can make the discretized actions unable to complete the task. In this case, do we need a number of tokens proportional to $2^{d_{act}}$?

---

> ### Author Response · Authors · 2023-11-15
> **Thanks for your review!**
>
> Thank you for your detailed review! We appreciate your critical perspective. We take your feedback seriously and believe it has already greatly improved the presentation! We use teal for revisions related to your comments.
>
> ### Strengths
>
> Thank you for acknowledging these aspects of our work. We are excited to see how far this fast tokenization idea goes in the future!
>
> ### Weaknesses
>
> > About paper-writing...
>
> Thank you for your comment! We wished to present the motivation from a broad perspective, as it is not always completely clear who the audience may be, and they may have different levels of experience with the subject matter, but we appreciate the precise perspective! We've made changes condensing the introduction and related work that hopefully answer these concerns.
>
> > About the observation-based distance score...
>
> We agree the presentation does not motivate the metric sufficiently. This metric is drawn from prior work (LSD eqn. 3, CSD eqn. 15) where such Euclidean and Mahalanobis distances were shown to have good properties in state-based domains with complex embodiment like robot manipulation (e.g. Fetch and Kitchen). We have changed the text to make this more clear. We agree that non-state-based domains may present an issue, which we discuss in Limitations, and ultimately believe using a visual latent space is the correct approach. Given prior work (DIAYN, LSD, CSD, LfP, SFP, SPiRL) began in state-based domains and sparse-reward is already difficult in state-based domains, we thought it best to tackle one problem at a time.
>
> > Soundness of experiments
>
> >> Lack a state-based baseline...
>
> It appears our presentation was not sufficiently clear to motivate these choices in the original text. We've made large changes to Section 4.1 and 4.3 to motivate our choices, and move experiments comparing to conditioned skills into the main text.
>
> The reason that we see the correct baseline as SSP and not SPiRL is that we extract skills without any information on when to use those skills, then learn an RL policy on top of those skills. SSP does the exact same with a VAE. SPiRL extracts both skills **and** the knowledge of when to use those skills in the environment, which leads to faster and more stable learning when the final task aligns well with the demonstration data, but handicaps learning in new environments. If we wanted to compare with observation-conditioned methods, the correct approach would be to design a prior that looks at the particular observations when our subwords are used, and biases exploration with that prior like SPiRL. As we did not implement such a method, it seems correct to compare directly against SSP. Still, we take this point into account and have added experiments in Section 4.3 with comparison to conditioned skills and their generalization behavior.
>
> >> Training curves of returns are usually required...
>
> We are happy to include training curves. We initially thought it unwise given all 0s that are reported in tables correspond to runs that have 0 reward the entire run, thus most of the content would be flat lines. Prior work OPAL also presents tables, so for space we thought it wouldn't be an issue. We agree that there is inherent instability in RL, which is exacerbated by sparse-reward settings. To give more information as to the actual behavior of methods regardless of reward, we include exploration in Figure 4 which supports our method. In Figure 5 we make the point that, even when reward is low, exploration is still very good, thus the issue with instability has more to do with optimization on the RL side. We have explored many existing tricks to fix this (e.g. Prioritized Replay, REDQ, RND, DDQN, extra entropy regularization, hyperparameter tuning, etc.) but were not able to achieve completely stable results. To further this analysis we include visualizations in Appendix J of a seed of one of our runs whose Q-function collapses, that achieves high reward in the middle of training and 0 at the end.
>
> > Lack the discussion of disadvantages of state-free skills...
>
> We agree that our presentation was likely too imbalanced. We've now made large changes to Section 4.3, including extra experiments, which discuss this in a more balanced fashion.
>
> ### Questions
>
> > ...introduce the topic of skill discovery more directly...
>
> Thanks for this pointer! We've added some discussion at the beginning of Section 3 that does so.
>
> > ...behavior in the dataset is not diverse?
>
> We've now included experiments in Appendix I looking at diversity of demonstrations in for Hopper, where D4RL provides a clear axis of quality. We also provide ablations in Appendix H for varying discretization. A large number of clusters is preferred, but it is still possible with a small number. If we believed every possible articulation was necessary to model, then an exponential number of tokens makes sense, but this may not be the case in many tasks.
>
> Thanks again for your detailed review! Looking forward to discussion!

---

> ### Author Response · Authors · 2023-11-21
> **Discussion period ending soon**
>
> Given the discussion period ends on November 22nd, we wanted to see if there were any further questions/comments that we might address related to the rebuttal. We've attempted to respond in detail to all your concerns, and would greatly appreciate your prompt response!

---

> > ### Comment · Reviewer_NHza · 2023-11-23
> >
> > Thank you for taking the time to respond to my questions with paper revision, additional results, and explanations. The paper writing in introduction and related work has been improved. But my initial concerns about other aspects still remain.
> >
> > - **Soundness of results:** I think your explanation of not providing RL training curves is questionable.
> >
> > "Prior work OPAL also presents tables". OPAL [1] is a **fully offline** method which trains policies on a **fixed dataset**, this type of algorithms can converge with lower variance and have no considerations in sample efficiency. However, your method uses **online RL** for downstream tasks. All other works [2,3] in this field report training curves.
> >
> > "there is inherent instability in RL, which is exacerbated by sparse-reward settings". But the RL training curves in SPiRL [2]
> > do not have large variance in the same sparse-reward tasks. If your results have large variance, this is may caused by the the action space made of state-free tokenized skills, rather than the issues from RL algorithms or sparse reward. Unfortunately, this issue cannot be discussed as the curves are not provided.
> >
> > "We initially thought it unwise given all 0s that are reported in tables correspond to runs that have 0 reward the entire run". I think the key issue here is not the all-0 results of some baselines, but **the extreme large variance** of your method (in Table 1, the std even exceeds the mean in some data). It is necessary to at least provide the curves in your Appendix.
> > And for situations where the variance is too large, I also have doubts about whether 5 seeds are enough.
> >
> > - **About state-free skills:**
> >
> > In your additional results in 4.3, "while SPiRL-cl fails on CoinRun, likely due to differences between the dataset for CoinRun (easy levels) and the downstream task (hard levels)". But your results of SPiRL in CoinRun (Table 4) is the best among all methods, showing that state-based SPiRL performs well with this domain gap in CoinRun.
> >
> > The weakness of the large variance of your method is not discussed, which is may caused by state-free skill learning. Intuitively, I think with state-free skills, the high-level RL policy can randomly choose many unreasonable skills given a state (e.g., choose to turn left on the straight path in Maze), increasing the instability of RL.
> >
> > "(state-based skills) may be detrimental when we do not have access to sufficiently general demonstrations, like the ~10,000
> > trajectories in randomized environments that SPiRL uses for visual PointMaze". But the problem setting of your paper and SPiRL is to learn skills from **task-free dataset**. Because it is easy to collect large task-free datasets, extracting diverse skills from it to accelerate RL in unknown tasks is promising. Learning from small data (from scratch) to generalize to domain-shift is itself an ill-posed problem.
> >
> > [1] OPAL: Offline primitive discovery for accelerating offline reinforcement learning. 2021
> >
> > [2] Accelerating Reinforcement Learning with Learned Skill Priors. 2021
> >
> > [3] Reinforcement Learning with Action-Free Pre-Training from Videos. 2022
> >
> > To this end, I keep my initial assessment on this paper.

---

> > > ### Author Response · Authors · 2023-11-23
> > >
> > > Thank you for responding. We are happy to hear that you find the writing in the introduction and related work to have improved.
> > >
> > > > OPAL [1] is a fully offline method which trains policies on a fixed dataset,
> > >
> > > This is not entirely correct. Like our method, OPAL learns skills ("primitives") from an offline dataset. However, also like our method, OPAL includes results that demonstrate the ability to learn a policy over these primitives via **online RL** (Section 5.3 and Table 3, see [https://arxiv.org/pdf/2010.13611.pdf](https://arxiv.org/pdf/2010.13611.pdf)). We reference OPAL's use of tables in particular because they are the only method that evaluates on similarly difficult sparse-reward domains (i.e., antmaze). However, as noted below and in our initial response, we would be happy to provide training curves.
> > >
> > > > But the RL training curves in SPiRL [2] do not have large variance in the same sparse-reward tasks.
> > >
> > > With the exception of Kitchen, the SPiRL paper does not include results on the same sparse-reward tasks. The four domains we choose are the most challenging domains taken from prior literature (the OPAL authors provide a nice discussion of the particular difficulties of antmaze [https://openreview.net/forum?id=V69LGwJ0lIN&noteId=gmdyGKpnMDy](https://openreview.net/forum?id=V69LGwJ0lIN&noteId=gmdyGKpnMDy)). SPiRL evaluates on Pointmaze, Block-stacking and Kitchen, the first two of which are much easier than CoinRun and Antmaze. We have results on Pointmaze-large and our method (along with SSP, SPiRL, SPiRL-cl) achieves perfect performance (we left the results out of the paper as we didn't find them particularly informative, but would be happy to include them). On more challenging domains (Table 4), **SPiRL exhibits large variance**: $0.40 \pm 0.49$ on antmaze-medium-diverse and $5.32 \pm 5.41$ on CoinRun. Meanwhile, SPiRL fails entirely on antmaze-large-diverse ($0.0$).
> > >
> > > >  If your results have large variance, this is may caused by the the action space made of state-free tokenized skills, rather than the issues from RL algorithms or sparse reward.
> > >
> > > The (i) high variance of SPiRL on two of the four domains (antmaze-medium-diverse and CoinRun) and failure on the third (antmaze-large-diverse), together with (ii) the high quality of the exploration of our method even for seeds that achieve zero reward (Figure 5) suggest that **the variance is not a result of the action space of our state-free tokenization**, but rather a result of the difficulty of the sparse-reward domains. This is consistent with the findings of Zhou et al. (2022). The perfect performance of our method on PointMaze-large referenced above further supports this conclusion.
> > >
> > > **Training Curves**: As we stated in our initial response, we are happy to provide the training curves for our method as well as the baseline. We were sincere the reasoning provided above---it is not our intention to obscure the behavior of our method. Unfortunately, we just received this response and the author-reviewer discussion period closes in two hours, while the author who has these plots is presumably asleep (it is the middle of the night).
> > >
> > >
> > > > The weakness of the large variance of your method is not discussed, which is may caused by state-free skill learning
> > >
> > > As noted above, SPiRL similarly exhibits high variance on two of the four domains (antmaze-medium-diverse and CoinRun) and completely fails on the third (antmaze-large-diverse), suggesting that the variance is not caused by state-free skill learning. Given the 0-1 nature of the reward, the high variance is a result of some seeds receiving 0 reward, but as we show in Figure 5, even these zero-reward seeds exhibit strong exploration behavior, and the goal of our paper is to improve exploration in RL.
> > >
> > > > But the problem setting of your paper and SPiRL is to learn skills from task-free dataset.
> > >
> > > Neither SPiRL nor our method are designed to learn from demonstrations that are completely disjoint from the target task. The point that we were trying to make with the referenced text is that in order to generalize to new tasks, state-conditioned methods require a large, diverse dataset (10k random trajectories in PointMaze for SPiRL) in order to mitigate the covariate shift that is inherent in state-conditioning. The authors of SFP
> > > (Bagatella et al., 2022, Figures 7, 16) draw similar conclusions. In many real-world tasks, particularly those that involve embodiment (e.g., robotics), it is not easy to collect large datasets.

---

### Official Review · Reviewer_AfCr · 2023-11-01

**Soundness:** 3 good
**Presentation:** 2 fair
**Contribution:** 3 good
**Rating:** 5
**Confidence:** 2

**Summary:**

This paper presents a method for discovering skills by first discretizing the action space through clustering, and then leveraging a tokenization technique borrowed from NLP to generate temporally extended action. The skills discovered by the proposed method supports better exploration that lead to better performance than baselines in sparse-reward settings.

**Strengths:**

This paper presents a novel idea for discovering skills borrowing the idea of tokenization from NLP. The learned skills from offline dataset provides rich exploration behavior for online RL. The proposed method is simple but effective, especially for sparse reward RL tasks. Experiments in the ant maze domain also demonstrate the generalization capability of learned skills.

**Weaknesses:**

One major weakness of the proposed approach is that discretization removes resolution from the action space, which can be detrimental in settings that require the full range of actions, such as fast locomotion or precise manipulation. Experiments demonstrating this failure mode would be useful for the audience to understand this limitation.
Additionally, execution of the identified skills is currently open loop, which can lead to inefficient and unsafe exploration.
At the same time, the merging process used in the approach may be too computationally expensive to perform in high-dimensional visual input domains.
The authors discuss these limitations and suggested potential solutions, but they still represent significant challenges to the practical application of the approach.

Skill discovery is important beyong RL settings, more experiments and discussions on applying tokenized skills for imitation learning tasks would also be interesting. Imitation-based methods should also be considered as baselines in the experiments.

The success of skill discovery seems heavily dependent on the provided dataset, it would be useful to vary the diversity of the offline dataset to test the limitations of the method.

**Questions:**

See concerns in weakness.

---

> ### Author Response · Authors · 2023-11-15
> **Thanks for your review!**
>
> Thank you for spending time reviewing our paper! We take your comments seriously and address them below. We use reddish-purple in the text for revisions based on your comments.
>
> ### Strengths:
>
> Thanks for noticing these aspects. We are personally fans of simple and fast methodology, and hope such an approach might lead to fast RL.
>
> ### Weaknesses
>
> > ...discretization removes resolution...
>
> Thank you for your detailed read of our limitations section. On the note of fast locomotion, we have included an experiment on Hopper with dense-reward using frequency-based subwords, where we see that greater number of clusters leads to greater scores, because of more resolution in the action space. We include this experiment in Appendix H.
>
> > ...execution of identified skills is currently open loop...
>
> Due to space it was difficult to discuss fixes to this in the limitations section, but following prior work (SPiRL, DIAYN, LSD, CSD), we though it best to begin in the open-loop setting. One straightforward fix to this issue would be to run a separate small neural network responsible for deciding termination. Still, we thought it best to leave this to future work given space.
>
> > ...too computationally expensive to perform in high-dimensional visual input domains...
>
> We wanted to begin in state-based domains as the sparse-reward problem is already quite difficult in such domains (notice how Kitchen scores are quite low for all), and is a starting point for many methods (DIAYN, LSD, CSD, SPiRL, LfP). We believe that in visual domains, working with visual feature space for measuring distances would be the way to go, and this wouldn't suffer from the computational expense that we discussed in Limitations. This is definitely a promising direction, but given time and space constraints we thought it best to leave to future work.
>
> > ...Skill discovery is important beyond RL settings...
>
> Thank you for this comment. We agree that applying tokenized actions to imitation learning is very interesting. So much so that we are already working on followup in this direction!
>
> > Imitation-based methods should also be considered as baselines...
>
> It appears our presentation was lacking, and we've made changes to the text as a result. We respectfully disagree that this is a correct comparison. The goal of our paper is to improve exploration in RL, not because these particular tasks cannot be solved by methods with many demonstrations, but because if we want to learn tasks for which we do not already have demonstrations, we will need to do exploration and RL. Our method is designed in such a way that if data is collected in a different task that shares the same action space, then skills can generalize (Table 4 and discussion at the end of Limitations). For imitation-like baselines, SPiRL and SPiRL-cl are observation-conditioned, which means they already have an association between the observations and actions. We see that these methods are best in a setting where the demonstrations cover the final task (Kitchen), but may suffer in other domains with less overlap (AntMaze, CoinRun). In the case where no additional exploration is allowed, imitation learning methods would struggle in the generalization setting. We believe that imitation learning methods have their place, but think they are somewhat orthogonal to exploration. Hopefully this is clearer, and we have made updates to Section 4.3 to clarify this point.
>
> > ...heavily dependent on the provided dataset...
>
> Unfortunately the sparse-reward tasks in D4RL do not have datasets of varying "quality" necessarily, only overlap with the final task, but we conduct an experiment with datasets of varying quality for the dense-reward Hopper task in Appendix I. Here we see that random transitions and expert-transitions extract skills with similar performance, while medium-transitions give much worse skills. This is a puzzling result until one investigates the discovered skills: expert transitions extract useable chunks, random transitions extract very short action sequences so it is easy for the policy to recombine them, and medium transitions find very long, suboptimal skills.
>
> Thank you again for your review, and looking forward to further discussion!

---

> ### Author Response · Authors · 2023-11-21
> **Discussion period ends soon**
>
> Given the discussion period ends on November 22nd, we wanted to see if there were any further questions/comments that we might address related to the rebuttal. We hope the changes to the text and the rebuttal are helpful, and we would greatly appreciate your prompt response!

---

### Official Review · Reviewer_mNUW · 2023-11-05

**Soundness:** 3 good
**Presentation:** 2 fair
**Contribution:** 3 good
**Rating:** 6
**Confidence:** 4

**Summary:**

This paper proposes using the recently popularised byte-pair encoding strategy for tokenising in NLP to tackle the exploration problem in deep RL. This involves the key steps of collecting a dataset of demonstrations, discretising actions, tokenising them, and then using this tokenised action space to learn a policy using any RL algorithm. Their approach demonstrates better performance on many different environments than some other RL and HRL baselines.

**Strengths:**

- The main strength of this paper is that this is a neat idea that borrows insights from another field (NLP in this case) to propose an innovative and intuitive solution to one of the main challenges in HRL.
- The results indicate that this approach shows better performance than the other baselines in several sparse-reward environments.
- The details regarding implementation of the approach and baselines, resources used, wall clock time, and hyperparameters provided in the Appendix are quite exhaustive and will greatly help with reproducibility.

**Weaknesses:**

- Algorithm 1 could be presented in a much clearer way by adding comments for each of the steps and what each of the variables means.
- I am unclear on why the mean vector for each subword is calculated in this way. Is there prior work that does it this particular way, or is it a design decision? Maybe this can be elaborated in the paper.
- I think some important ablations should have been included in the experiments:
    - Clustering-only: how far can you go with pure discretisation of the action space, without learning skills?
    - Alternate tokenisation strategy: if instead of having a preference for longer skills/subwords, what would the performance look like if the size of each skill was fixed? ($N_{max}$ seems to have been varied across environments, per the appendix, so this would also be an interesting ablation).
- The description of the type of data used in each environment is missing in the main text (it is present in the appendix). Given that the appendix mentions that the ant maze data consists of poor samples, whereas the kitchen data consists of nearly perfect demonstration, this design decision warrants an explanation/analysis in the main text.
- There is some explanation in the appendix regarding the choice of vocabulary size for the different environments, but it is tied to the choice of data used for tokenisation. It would be nice to see a separate ablation experiment for this hyperparameter.

**Questions:**

- The paper mentions that for frequency-based merging, the longest subword is selected and all its constituent subwords are pruned. My understanding is that this biases the approach in favour of longer subwords/skills. Could you confirm?
- From Table 1, it appears that frequency-based merging has little to no impact on performance; the key gains seem to be coming from distance-based merging. Since distance-based merging entails knowledge of the environment in that we want agents to cover longer distances through their actions, is it fair to compare this approach with the other baselines (SSP/SFP/SAC)?
- Why did the kitchen dataset consist of expert demonstrations, when the other datasets did not?
- Did you perform any analysis/have any insights on the frequency of usage of each skill for a trained policy?

---

> ### Author Response · Authors · 2023-11-15
> **Thanks for your review!**
>
> Thank you for spending time with the paper! Your comments are greatly appreciated and we will try to respond in full below. We use orange in the text for revisions corresponding to your comments.
>
> ### Strengths
>
> Thanks for making these notes. We are excited at the prospect of porting a simple technique which can hopefully lead to faster RL in the long run.
>
> ### Weaknesses
>
> > Algorithm 1 could be presented...
>
> We agree that the presentation is much too condensed, and have added comments for important steps in the revised version!
>
> > I am unclear on why...
>
> It is true that the presentation is quite terse. For this choice we are inspired by prior work LSD and CSD (equation 15). We have added description to the corresponding section that hopefully better motivates.
>
> > ...some important ablations...
> >> Clustering-only...
>
> It appears our communication was not as clear as it could have been. The clustering-only ablation is already in the paper and corresponds to the SAC-discrete column in Table 1, where clustered actions are used as discrete actions. We can see the poor exploration behavior in Figure 4a. We've added small changes to the text to clarify this.
>
> >> Alternate tokenisation strategy...
>
> We varied hyperparameters across environments in order to get skills of length close to 10, to be comparable to baselines. Our goal was to test the viability of the simplest possible tokenization strategy available, hence why we began with BPE, a quite standard algorithm. We also discuss some shortcomings of grammar-learning methods like Sequitur in the end of the related work. Are you suggesting a version of BPE with an additional cutoff? It is not trivial to tightly control the length of skills BPE discovers.
>
> > The description of the data...
>
> Thanks for the suggestion, we've made small changes to the main text with more details. Our data is inherited from D4RL, a standard set of datasets and tasks in the community. We did not collect this data, and thought it best to work in a setting that readers might be familiar with.
>
> > ...choice of vocabulary size...
>
> This is a good suggestion, and it seems our initial presentation in Appendix A was not clear. Do those experiments satisfy your questions? In the main text we make choices of hyperparameters such that we are comparable to baselines, as if our skills were shorter or longer there would be a confounding factor.
>
> ### Questions
>
> > ...biases the approach in favour of longer subwords/skills...
>
> Yes this is true. For more context, the maximum length of subwords ultimately is decided by the maximum number of merges allowed, which we set such that our subwords are on average the same length as the baseline SSP (10 steps). If we allowed more merges, the subwords would be even longer. Alternatively one could redesign BPE for this setting, but we wanted to start with the simplest choice.
>
> > ...frequency-based merging has little to no impact...
>
> We would like to gently point out that frequency-based merging is comparable except in AntMaze domains, due to the suboptimality of demonstrations. We design this additional change, inspired by work like DIAYN (Section 4.2.2) that introduces priors with knowledge of the observation space structure, in order to make the argument that a tweak makes our method viable. Perhaps this is not presented the best and we are happy to revisit!
>
> > Why did the kitchen dataset consist...
>
> Thanks for the question, as in our answers above we used a standard dataset D4RL, whose collection we do not have control over. We thought this would be a better choice than bespoke datasets.
>
> > ...frequency of usage of each skill...
>
> With regards to frequency of skill usage we do not have this information, but would be happy to include such an analysis.
>
> Thanks again for your many questions and detailed reading, looking forward to more discussion!

---

> ### Author Response · Authors · 2023-11-21
> **Discussion period ending soon**
>
> Given the discussion period ends on November 22nd, we wanted to see if there were any further questions/comments that we might address related to the rebuttal. We would greatly appreciate your prompt response!

---

> > ### Comment · Reviewer_mNUW · 2023-11-22
> >
> > Thank you for your answers to my questions. I am choosing to maintain my score. A few points (score-unrelated) below:
> >
> > - The clarification on the data used was helpful; my main point was that in spite of the datasets being standard datasets used by the community, it would be helpful to talk about the data in the context of your proposed approach; how might using a certain type of data (expert demonstrations, suboptimal demonstrations) affect the efficacy of your proposed approach? That might make a stronger case for using your approach in certain situations more than others.
> > - Adding comments to Algorithm 1 was helpful, but my overall concern still remains; comments such as "compute vectors" are not as helpful as if those lines were replaced with understandable pseudocode.
> > - My comment on an alternate tokenisation strategy ties in with my questions on longer subwords being discovered, and analysis of usage of each skill; an analysis would give statistics on the lengths of skills and their usage, thereby determining whether an explicit cutoff for BPE skills is necessary or not.

---

> > > ### Author Response · Authors · 2023-11-22
> > > **Thanks for the response**
> > >
> > > Thanks for your continued engagement! We are happy to see that we've satisfied many issues, and we address lingering ones below:
> > >
> > > > The clarification on the data used was helpful...
> > >
> > > The original weakness we responded to was
> > > >> Given that the appendix mentions that the ant maze data consists of poor samples, whereas the kitchen data consists of nearly perfect demonstration, this design decision warrants an explanation/analysis in the main text,"
> > >
> > > from which we understood that we were to clarify why we chose these datasets, and we chose these datasets because they are the nontrivial sparse-reward tasks in an existing benchmark (D4RL). As to comment on how the quality affects the method, which we did not understand was being requested, perhaps this is not clear in the text already, but it is referenced in a few places:
> > > - beginning of Section 3.3 as motivation for alternate merging:
> > > >> In NLP, we often have access to a large amount of text data from (mostly) correct human authors. However, for robotics applications we may not have the same quantity of near-optimal (or even suboptimal) demonstrations. As a result, it may be undesirable to merge tokens based on frequency alone.
> > > - Section 4.1, commenting on Table 1:
> > > >> Failures of frequency-based merging in AntMazes are directly attributable to the discovery of long, constant sequences of actions, likely due to suboptimal demonstration trajectories that often jitter in place
> > > - Limitations paragraph:
> > > >> We also speculate that higher-quality demonstrations could allow us to generate skills simply by merging based on frequency (Table 1, CoinRun), and these demonstrations may be easy to obtain if they don’t need to be collected in the deployment domain (Table 5).
> > >
> > > If there is a way you believe this could be made more precise, we are happy to make changes!
> > >
> > > > Adding comments to Algorithm 1 was helpful...
> > >
> > > The text in Section 3 is meant to provide the high level view, where the algorithm provides all details in one place for precision. We're happy to provide a pseudocode implementation and defer the exact algorithm to the Appendix if that is preferred. The initial feedback asked for comments
> > > >> Algorithm 1 could be presented in a much clearer way by adding comments for each of the steps and what each of the variables means.
> > >
> > > and our current change was a difficult balance of space constraints to keep comments on a single line, with more detailed description.
> > >
> > > > My comment on an alternate tokenisation strategy ties...
> > >
> > > We agree that such analysis can be interesting. Anecdotally it is not the most informative. A policy that is successful on antmaze-medium uses mostly a single skill of length 9, while interspersing a few other skills of length 8, 17 and 20. A policy that is unsuccessful only uses skills of length 4. More informative results might include examining training dynamics of skill usage (difficult to do before the end of the discussion period), or qualitative labels for each skill, but those are manually intensive to collect.
> > >
> > > Does you have a specific, fixed-length alternate tokenization strategy in mind? Sequitur (discussed at the end of the related work) will have similar issues. Unigram and Wordpiece will also behave like BPE, with variable lengths. It is not clear whether an explicit cutoff is meaningful or useful (certain motor routines may have certain natural "lengths"), though it will certainly come with tradeoffs.
> > >
> > > Thanks again for your detailed responses.

---

### Author Response · Authors · 2023-11-15
**Thanks to all for their reviews!**

Thank you to all reviews for their time and detailed criticisms! We believe the paper has been substantially improved through this feedback.

For relevant changes to each reviewer we highlight text in different colors:
- mNUW: orange
- AfCr: reddish purple
- NHza: teal
- Y696: dark purple

### (Non-exhaustive) changelog
- Condensed introduction to shorten discussion of loosely related methods (NHza)
- Shortened text relating to language models in introduction (NHza)
- Removed related work section on skill learning from interaction (NHza/Y696)
- Added small clarification to beginning of method section with problem setting
- Added comments to algorithm statements (mNUW)
- Named our method (SaS-freq, SaS-dist) (SaS for Subwords as Skills) in Section 3.3
- Clarified motivation for merging function in Section 3.3 (mNUW, NHza)
- Added transition to opening of Section 4 to clarify problem setting (NHza, Y696)
- Renamed Section 4.1 and rewrote transition (NHza)
- Removed extra related work discussion in Section 4.1 (NHza)
- Separated Task and Baseline paragraphs in Section 4.1 (Y696)
- Move results in Appendix G on observation-conditioned skills into Section 4.3 (mNUW, NHza, Y696)
- Discussed observation conditioning tradeoffs in Section 4.3 and further clarified differences of assumptions to our setting (NHza, Y696)
- New experiments on discretization in locomotion in Appendix H (AfCr)
- New experiments on data quality in Appendix I (NHza, AfCr)
- Q-collapse visualizations to further support conclusions of Figure 5 in Appendix J (NHza)

### Method acronyms:
Method acronyms that are referred to across reviews are given below:
- SPiRL: Pertsch et al. 2020. Accelerating reinforcement learning with learned skill priors
- SFP: Bagatella et al. 2022. SFP: State-free priors for exploration in off-policy reinforcement learning
- DIAYN: Eysenbach et al. 2018. Diversity is all you need: Learning skills without a reward function
- LSD: Park et al. 2022. Lipschitz-constrained unsupervised skil discovery
- CSD: Park et al. 2023. Controllability-Aware Unsupervised Skill Discovery
- LfP: Lynch et al. 2020. Learning latent plans from play

Looking forward to discussion!

---

### Meta-Review · Area_Chair_GU5n · 2023-12-05

**Metareview:**

**Summary**: This paper proposes a method for skill discovery that discretizes the action space through clustering, then leverages a tokenization technique borrowed from NLP to generate temporally extended actions. The method is evaluated in AntMaze, Kitchen and CoinRun and is shown to outperform some state-free skill learning baselines and vanilla SAC.

**Strengths**:
- Leveraging tokenization from NLP for skill discovery is a novel and creative idea.
- The results show that this approach outperforms baselines in several sparse-reward environments.
- The evaluation looks at several aspects of the proposed approach beyond performance: computation efficiency, exploration behavior, domain generalization.
- The appendix contains many implementation details and hyperparameters that significantly aid reproducibility.

**Weaknesses**:
- There are several writing clarity issues that seem to only be partially addressed in the rebuttal. For instance, I thought the introduction spent too much time on related work and background and not enough time describing the proposed approach. One reviewer felt similarly ("In Introduction, the background of RL and exploration is too long"), yet still this was not addressed in the new draft. The front figure is not very descriptive and wastes a lot of white space. The tables are hard to read, which made it difficult for me as a reader to draw conclusions about the method. Overall, there are several serious issues with writing presentation and clarity that would need to be resolved in a future version. I am not certain that I trust these writing issues could be resolved for a potential camera ready version.
- A reviewer takes issue with the author's refusal to provide training curves, which seems to be a standard practice in online RL papers. The reviewer is skeptical of the soundness of the results due to the apparent high variance of the method.

Overall, most reviewers seem on the fence about the paper, either slightly in favor of acceptance (two 6 scores) or slightly in favor of rejection (a 5 score). One reviewer is confidently for rejection and provides reasons for keeping their score despite the rebuttal (albeit those reasons came close to the discussion deadline). In my opinion, most of the problems plaguing this paper stem from the writing and presentation: 1. the introduction is mostly background, so as readers we don't get to fully appreciate the novelty and contribution of this approach; 2. given that, the related work section is much too long and several paragraphs could've been trimmed to allow for more space for describing the experimental results; 3. the experimental results are difficult to parse and understand (particularly the tables), so again, as readers we can't fully appreciate the significance of the approach and its results. I believe these presentation issues are severe enough that the paper cannot be published in its current form, despite exhibiting several positive results.

**Justification For Why Not Higher Score:**

To reiterate my reasoning above, I believe the paper suffers from severe clarity and presentation issues that make it difficult to judge the contributions and significance of results. I am recommending rejection but would not mind if the score would be bumped up to acceptance with poster.

**Justification For Why Not Lower Score:**

N/A

---

### Decision · Program_Chairs · 2024-01-16

Reject